

# Drought increased since the mid-20th century in the northern South American Altiplano revealed by a 389-year precipitation record

Mariano S. Morales[1,2], Doris B. Crispín De La Cruz[1], Claudio Álvarez[3,4,5], Duncan A. Christie[3,5,6], Eugenia Ferrero[1,2], Laia Andreu-Hayles[7,8,9], Ricardo Villalba[2], Anthony Guerra[10,11], Ginette Ticse-Otarola[1], Ernesto Rodríguez-Ramírez[1], Rosmery LLocclla Martínez[1], Joali Sanchez-Ferrer[1], Edilson J. Requena-Rojas[1]

[1]Laboratorio de Dendrocronología, Universidad Continental, Huancayo, 12000, Perú
[2]Instituto Argentino de Nivología, Glaciología y Ciencias Ambientales, CONICET, Mendoza, 5500, Argentina
[3]Laboratorio de Dendrocronología y Cambio Global, Instituto de Conservación Biodiversidad y Territorio, Universidad Austral de Chile, Valdivia, 5110566, Chile.
[4]Escuela de Graduados, Facultad de Ciencias Forestales y Recursos Naturales, Universidad Austral de Chile, Valdivia, 5110566, Chile.
[5]Center for Climate and Resilience Research (CR)[2], Santiago, 9160000, Chile.
[6]Cape Horn International Center (CHIC), Punta Arenas, 6200000, Chile.
[7]Lamont-Doherty Earth Observatory of Columbia University, New York, NY10964, United States
[8]CREAF, Bellaterra (Cerdanyola del Vallés), Barcelona, 081093, Spain.
[9]ICREA, Pg. Lluís Companys 23, Barcelona, 08010, Spain.
[10]Facultad de Ciencias Forestales y del Medio Ambiente, Universidad Nacional del Centro del Perú, Huancayo, Perú, Avenida Mariscal Castilla 3909, Huancayo, 12006, Perú.
[11]Missouri Botanical Garden, Pasco, Oxapampa, 19230, Perú.

*Correspondence to*: Mariano S. Morales (mmorales@mendoza-conicet.gob.ar)

**Abstract.** Given the short span of instrumental precipitation records in the South American Altiplano, long-term hydroclimatic records are needed to understand the nature of climate variability and to improve the predictability of precipitation, a key natural resource for the socio-economic development in the Altiplano and adjacent arid lowlands. In this region grows *Polylepis tarapacana*, a long-lived tree species that is very sensitive to hydroclimatic changes and have been widely used for tree-ring studies in the central and southern Altiplano. However, in the northern sector of the Peruvian and Chilean Altiplano (16º-19º S) still exist a gap of hydroclimatic tree-ring records. Our study provides an overview of the temporal evolution of annual precipitation for the period 1625-2013 CE at the northern South American Altiplano, allowing for the identification of wet or dry periods based on a regional reconstruction composed by three *P. tarapacana* chronologies. An increase in the occurrence rate of extreme dry events, together with a decreasing trend in the reconstructed precipitation, have been recorded since the 1970s decade in the northern Altiplano within the context of the last ~four centuries. The average precipitation of the last 17-year stands out as the driest in our 389-years reconstruction. We revealed a temporal and spatial synchrony across the Altiplano region of wet conditions during the first half of the 19th century and the drought conditions since mid 1970s recorded by independent tree-ring based hydroclimate reconstructions and several


paleoclimatic records based on other proxies available for the tropical Andes. The rainfall reconstruction provides also valuable information about the ENSO influences in the northern Altiplano precipitation. The spectral properties of the rainfall reconstruction showed strong imprints of ENSO variability at decadal, sub-decadal and inter-annual time-scale, in

particular from the Pacific N3 sector. Overall, the remarkable recent reduction in precipitation in comparison with previous centuries, the increase in extreme dry events and the coupling between precipitation and ENSO variability reported by this work is essential information in the context of the growing demand for water resources in the Altiplano that will contribute to a better understanding of the vulnerability/resilience of the region to the projected evapotranspiration increase for the 21st century associated to global warming.

**1. Introduction**

The Altiplano in the Central Andes (~16°-24° S) is the largest semi-arid high-altitude region in South America. With an average elevation of 4000 m a.s.l. and a large number of volcanoes up to 6,700 m a.s.l., the Altiplano has hosted the settlement of diverse communities over thousands of years. Historically, human activities in the Altiplano have been strongly modulated by hydroclimatic variations (Tandeter, 1991; Binford et al., 1997; Nielsen, 1999; Núñez et al., 2002). Water

resources in this dry environment are crucial for agriculture, wetlands for livestock breeding, mining and other socioeconomic activities. Extreme drought events in the Altiplano, such as those that occurred in 1982/83 and 1988/89, caused serious problems in freshwater supplies for population drinking water, local agricultural and livestock activities causing large economic losses (Tandeter, 1991; García et al., 2003, 2007; Buytaert and De Bièvre, 2012).

Precipitation in the Altiplano is highly seasonal being associated with the development of the South American summer

monsoon (SASM). The SASM presents a seasonal cycle including the onset (October-November), maturity (December-February) and die-off (March-April) periods, and is the dominant mode of climate variability over tropical South America being responsible for more than 60% of the total annual precipitation in its core region (Vera et al., 2006; Garreaud et al., 2009; Marengo et al., 2012; Vuille et al., 2012).

At the Altiplano, temperatures series consistently show a positive warming trend since the second half of the 20th century

(Lavado-Casimiro et al., 2013; Vuille et al., 2015; Hunziker et al., 2018), while trends in precipitation are weaker and show a high spatial heterogeneity largely determined by the complex Andean topography (Vuille et al., 2003; Bennett et al., 2016; Imfeld et al., 2021). For instance, Huerta and Lavado-Casimiro (2021) indicated no significant trends towards wet or dry conditions for total annual precipitation in the Peruvian Altiplano during the period 1971-2013. A declined, albeit non significant, was found for the period 1980-2010 in most of the stations located in the drier southern Altiplano in Bolivia and

Chile by Bennett et al. (2016). Similar results were described by Vuille et al. (2003) for the period 1950-1994. However, an increasing trend of total precipitation (austral summer) only for the northeast of the Peruvian Altiplano was also reported (Bennett et al., 2016). Imfeld et al. (2021) detected a positive trend in summer precipitation from the southern Andes of Peru



for the period 1965/66–2017/18. Similarly, Segura et al. (2020) founded a positive trend in the regional precipitation for the southern Tropical Andes (including the Altiplano) during the period 1982-2018 using the CHIRPS satellite product.

The climate projections based on global and regional circulation models under different greenhouse gas (GHG) emissions scenarios show a steady warming for the Altiplano throughout the 21st century (Bradley et al., 2006; Urrutia and Vuille, 2009; Seth et al., 2010; IPCC, 2021). Summer precipitation is projected to decrease to the end of the 21st century, posing serious challenges for future water supply in this semi-arid region (Urrutia and Vuille, 2009; Minvielle and Garreaud, 2011; Neukom et al., 2015). Presently, due to the steady increase of human demand on water resources in the Central Andes it is a

key issue to identify the spatial patterns of precipitation, its temporal evolution, the return time interval of extreme hydroclimatic events and their future projections. The scarcity and short length of precipitation records in this region, usually less than 50 years, hampers the assessment of the range of natural hydroclimate variability and the evaluation of possible current changes. As an alternative, tree rings are one of the best paleoclimate proxies for reconstructing climate variability on a multi-century scale due to their annual resolution, high spatial coverage, and sensitivity to annual or seasonal climate

variations (Jones et al., 1998).

The tree species *Polylepis tarapacana* (Rosaceae) grows on the slopes of high volcanoes located between 4200-5200 m a.s.l. along the Altiplano, being adapted to cope with extreme dry conditions, low temperatures and high solar irradiance and can reach ages up to 700 years (Morales et al., 2004; Christie et al., 2009; Solíz et al., 2009; Garcia-Plazaola et al. 2015). This excellent dendrochronological properties of this species have been used to reconstruct the hydroclimatic evolution of the

central and southern regions of the South American Altiplano, as for example precipitation variability for the last 707 years from the Bolivian and Chilean sector (Morales et al., 2012) and surface variations of the Vilama-Coruto lake system for the last 601 years from northwestern Argentina to southwestern Bolivia (Morales et al., 2015). However, at present there is still a lack of long-term tree-ring chronologies to infer hydroclimatic variability in the northern sector of the Altiplano across the Peruvian and Chilean Andes spanning from 16º to 19º S. The main goal of this study is to develop a new tree-ring based

precipitation reconstruction for the northern sector of the Altiplano (16º-19º S), using a new set of *P. tarapacana* tree-ring chronologies. This newly high-resolution hydroclimate reconstruction will allow characterize the temporal variability of droughts and pluvials in this Andean sector over the past centuries and to evaluate and compare its evolution with southern regions in the Altiplano. Developing new high-resolution hydroclimate reconstructions is essential to understand the spatial variations in the Altiplano hydroclimate in a multi-century context, as well as to understand the influence of major drivers of

climate variability, such as El Niño-Southern Oscillation (ENSO) and SASM, in local precipitation across the entire Central Andes.

## 2. Data and Methods

### 2.1. Study area

The study region is across the Andes in southern Peru and northern Chile, also knows as northern Altiplano. The climate is

semi-arid with dry-cold winters and rainy-warm summers, with a total annual precipitation that range between 290-400 mm,



occurring >85% during summer (December-March). Maximum monthly temperatures are 12.9° C in July and 17.6° C in November, whereas the minimum temperature are -9.4° C in August and 1.7° C in January.

Two new chronologies of *P. tarapacana* were developed at the Chiluyo (CHI; Crispin-DeLaCruz et al., 2022) and Paucarani (PAU; Requena-Rojas et al., 2021) sites in the department of Tacna in Peru, and a third one at the Suriplaza site (SUR) in the
Arica-Parinacota region in Chile (Fig. 1; Table 1). None of the chronologies have been used in tree-ring based precipitation reconstruction before. Trees were sampled in steep slopes, rocky and xeric environments (Fig. 1b-d). Due to the presence of twisted stems and eccentric radial growth, cross-sections from *P. tarapacana* were taken from multi-stemmed living trees. Subfossil wood samples were also collected.

### 2.2. Tree-ring width chronologies development

Cross-sections were mounted and sanded following standard dendrochronological techniques (Stokes and Smiley, 1968) and each growth ring was visually dated following the Schulman's convention (1956) for the Southern Hemisphere that assigns to each ring the calendar year in which tree radial growth begins. Tree rings were measured with a binocular stereoscope to the nearest 0.001 mm. The COFECHA program (Holmes, 1983) was used to assess the quality of the visual dating and to identify measurement errors. Correctly dated ring-width measurements series were standardized to eliminate age-related
growth trends and minimize non-climatic related growth variations (Fritts, 1976).

We used a conservative method of standardization fitting negative exponential or linear curves with zero or negative slope to each individual tree-ring series. When the age-size trend is removed, some of the variance related to the climate signal can also be lost, leading to a trend distortion in the resulting index series (Melvin, 2004). To avoid this, the three standard chronologies were developed using the Signal Free procedure (RCSigFree program; Melvin, 2004; Melvin and Briffa, 2008;
Cook et al., 2017).

Previous studies with *P. tarapacana* chronologies across the central-south Altiplano have determined a high regional common signal among records highlighting a common spatial pattern at regional scale (Solíz et al. 2009, Morales et al., 2012, 2015; Crispin-DelaCruz et al. 2022). We evaluate this in our study region by a correlation matrix among the three signal free standard tree-ring chronologies over their well-replicated common period 1880-2007. Based on the highly
significant correlation coefficients that range from $r = 0.59$ to $0.67$ (n = 128, p < 0.001; Table 1), we created a regional well-replicated tree-ring chronology by assembling in a single record the 132 tree-ring width individual series from the three sites listed in Table 1. A correlation analysis of the three individual chronologies with the resulted regional chronology showed high and similar correlations ranging from 0.78 to 0.84 over the period 1880–2007 (Table 1).

### 2.3. Climate data collection and analysis

Monthly precipitation records were collected from seven meteorological stations located above 3200 m. a.s.l. in the northern Altiplano between 16° and 19° S. Only station records with less than 10% of missing data were considered (Table 2), and missing data from individual stations were estimated using linear regressions (Ramos-Calzado et al., 2008). We developed a



regional record of precipitation as follows: correlation coefficients between the standard regional chronology of *P. tarapacana* and the monthly variations of precipitation were used to define the best seasonal precipitation period related to

radial growth, resulting the November to January (NDJ) period (late spring - mid summer). Inter-annual time series of the total NDJ precipitation were created for each one of the seven available stations. Finally, the precipitation data from each station was normalized (z-score; Jones and Hulme, 1996) and averaged to obtain a regional index of precipitation from the northern South American Altiplano for the period 1970-2019. To assess the partial influence of temperature on tree growth we compiled monthly temperature data from nine high altitude (>3200 m. a.s.l.) meteorological stations from the Altiplano

(Table 1). Based on these data we computed a regional mean NDJ temperature index for the period 1963-2014 from the study region, following the same procedure described above used to develop the regional precipitation index.

### 2.4. Reconstruction method

We developed the annual precipitation reconstruction by regressing the regional standard chronology against NDJ precipitation utilizing a Principal Component regression approach (Cook et al., 2007). Predictors for the reconstruction

included the regional tree-ring chronology at time t+0 and its temporal lags (t+1 and t+2) significantly correlated ($\alpha = 0.05$) to the regional precipitation record during the 1970-2010 calibration period. This approach for selecting the predictors allows for a 3 years response to climate in the tree-ring width chronology (Fritts, 1976). These three lags were considered as candidate predictors of annual precipitation. As the predictors are intercorrelated, they were converted to orthogonal variables to reduce the dimension of the regression problem and to enhance the common precipitation signal, using a

Principal Component Analysis (Cooley and Lohnes, 1971). The selection criterion for choosing the best reconstruction model was that basing in maximizing the adjusted $R2$ ($R2$adj) by a stepwise multiple regression approach (Weisberg, 1985). Given the relative short precipitation record for calibration, the reconstruction model was developed using the leave-n-out cross-validation procedure, where n is five (1+2* number of positive lags: t+1 and t+2) (Michaelsen, 1987; Meko, 1997). In this approach five observations were successively withheld, a model was estimated on the remaining observations and a

prediction was made for the omitted observations. This resulted in a set of estimated precipitation values, which were compared to the observed precipitation observations to compute validation statistics of model accuracy and error. The leave-five-out analysis was performed using the GEOSA package in Matlab program (Dr. David Meko, Laboratory of Tree Ring Research, University of Arizona). To evaluate the similarities between the precipitation estimated by the reconstruction model and the observed instrumental data we used the proportion of variance explained by the regression (R2), the F-statistic

of the regression, and the autocorrelation in the regression residuals measured by the significance of the linear trend and the Durbin-Watson test (Draper and Smith, 1981). We also calculated the reduction of error (RE) during the verification period (Gordon, 1982), as well as the root mean square standard error (RMSE) as a measure of the inherent uncertainties in the reconstruction (Weisberg, 1985). The reconstructed precipitation values are anomalies expressed as percentages with respect to the common instrumental precipitation period 1982-2013.



## 2.5. Analysis of precipitation extreme events

A nonparametric Gaussian kernel function was applied to estimate the temporal changes in the occurrence-rates of extremes wet/dry annual events for the precipitation reconstruction (Mudelsee et al., 2003). The Kernel function detects non-monotonic trends and imposes no restrictions on the parameters. Here, the long-term changes in the probability of occurrence of one specific extreme event were explored by using a Kernel smoothing over a 40-y bandwidth. By resampling the list of yearly events in a 1000 bootstrap simulation we created confidence bands at the 95% level for a better interpretation of the estimations (Cowling et al., 1996; Mudelsee, 2014). The intensity of extreme wet and dry annual events was selected from values above or below the 95th and 5th percentiles, respectively.

### 2.6. Spatial patterns and spectral properties of the NDJ precipitation

To identify the spatial extent of the northern Altiplano precipitation across South America, we developed field correlation maps between the observed and the reconstructed NDJ precipitation with the 2.5° x 2.5° NDJ precipitation grid from the Global Precipitation Climatology Project database (GPCP; Adler et al., 2018). Field correlation maps were calculated for the common period 1979-2013 between the GPCP NDJ precipitation grid and the instrumental and reconstructed precipitation (https://psl.noaa.gov/data/correlation/).

Periodicities in the precipitation reconstruction were determined by means of spectral analysis. First, we establish the significant oscillatory modes in the instrumental and reconstructed precipitation using a Blackman–Tukey (BT) spectral density analysis (Jenkins and Watts, 1968) for the period spanning each time series. A 30% of the series length was used to determine the number of lags used for the auto- and cross-covariance functions. A Hamming filter was used to smooth the spectral functions and the confidence level of the spectrum (c.l. 95%) was computed from a first-order Markov null continuum based on the lag-1 autocorrelation (red noise) of the instrumental and the reconstructed time series (Mitchell et al., 1966). Second, we applied singular spectral analysis (SSA; Vautard and Ghil, 1989) to detect and extract the temporal patterns of the main oscillatory modes of reconstructed and instrumental precipitation time series. The SSA is a nonparametric method based on the empirical orthogonal function analysis that samples lagged copies of time series at equal time intervals and decomposes the instrumental and reconstructed series into oscillatory modes through the calculation of the eigenvalues and eigenvectors of the autocovariance matrix (Vautard and Ghil, 1989; Vautard, 1995). In this way SSA extracts the waveforms or reconstructed components of the main periodic or quasi-periodic modes of the analyzed time series and allows the assessment of the changes in the amplitude and phase of each mode over the time period of the series. We used the program AnClim (Štepánek, 2008) to perform the BT analysis and the SSA program (Boninsegna and Holmes, unpublished operating manual [on file at Laboratory of Tree-Ring Research, University of Arizona]) to extract the main oscillatory modes.





**2.7. ENSO signals in the northern Altiplano precipitation**

To asses the influence of the tropical Pacific sea surface temperatures (SSTs) on the reconstructed NJD precipitation variability over the northern Altiplano, we computed spatial correlations maps between the reconstructed NJD precipitation and the mean October-September SSTs from the gridded NCEP reanalysis global database (2.5º x 2.5º grid point resolution; Kistler et al., 2001) for the period 1949-2013.

We also correlated the inter-annual variations of the NDJ precipitation reconstruction with the mean annual (Oct-Sep) SSTs anomalies averaged for the NIÑO 3 sector (SSTs_N3) from the tropical Pacific (5º N – 5º S, 90º W – 150º W l) for the period 1870-2013. Finally, we extracted and compared the main oscillatory modes of variability of the precipitation reconstruction and the SSTs_N3 for the period 1870-2013. For that purpose, we performed singular spectral analyses (SSA; Vautard and Ghil, 1989) describe in section 2.7.

**2.8. Comparisons with hydroclimatic tree-ring based reconstructions from the Altiplano**

To evaluate similarities and differences of our NDJ precipitation reconstruction with independent tree-ring based hydroclimatic reconstructions from the South American Altiplano, we compared it with: (1) a precipitation reconstruction covering the period 1300-2009, based on *P. tarapacana* chronologies from the central and southern sectors of the Altiplano (Morales et al., 2012); (2) a tree-ring reconstruction of the variations in lake surface area from the Vilama-Coruto system in
the southwestern Altiplano covering the period 1407-2007 CE (Morales et al., 2015), and (3) a gridded reconstruction of the Palmer Drought Severity Index (scPDSI) covering the entire Altiplano region for the period 1400-2019 CE (Morales et al., 2020).

**3. Results**

**3.1. Regional tree-ring chronology and climate growth relationship**

The newly-developed regional "signal free" standard *P. tarapacana* chronology covers the period 1602-2015, but is well replicated with more than 10 ring-width series since the year 1625 (Fig. S1). In agreement to the sample size, the R-Bar and EPS statistics indicate a strong common signal among the individual timeseries that composed the regional chronology with a R-Bar mean value of 0.36 and an EPS value above the 0.85 threshold during the period 1625-2015 (Fig. S1; Table 1). A mean sensitivity value of 0.33 indicates that the *P. tarapacana* regional chronology contains relatively high inter annual
variability in radial growth (Table 1).

The radial growth variation of *P. tarapacana* is positively correlated (r = 0.63; p < 0.001) with the regional NDJ precipitation index of the previous growing season (Fig. 2a) and negative correlated (r = -0.53; p < 0.001) with the regional NDJ temperature index of the previous growing season (Fig. 2b) for the period 1970-2010. While prior late-spring to mid-



summer precipitation enhanced *P. tarapacana* growth response, temperatures for the same months exerted a negative effect

on tree growth.

**3.2. Reconstruction model and temporal evolution of the precipitation**

Our tree-ring reconstruction explains 42.5% of the total NDJ precipitation variability during the calibration period (1970-2010; Fig. 3a). The consistency between observed and reconstructed precipitation indicates a good quality of the calibration model to reproduce past precipitation variability for the study region. The predictive power of the calibration model is

consistent with a significant F value (F = 9.1; $p < 0.0001$) and a positive reduction of error (RE = 0.2). The residuals of the regression models have a normal distribution and were not significantly autocorrelated according to Durbin-Watson tests (Fig. 3b).

Our reconstruction spans the period 1625-2013, covering that last 389 years of NDJ precipitation variations from late spring-mid summer for the northern region of the Altiplano (Andean region from northern Chile-southern Peru; Fig. 3c). The record

is characterized by strong inter-annual variations that occur within decadal, multi-decadal and centennial fluctuations. Severe and persistent droughts and pluvials are identified in the reconstruction. During the period 1625-1661 rainfall variations oscillate around the long-term mean with the occurrence of three extreme wet years (1642/43/45). The period 1664 to 1724 was characterized by relative humid conditions interrupted by two extreme two-years dry events in 1692/93 and 1722/23. A persistent 15-year period of severe droughts conditions at the end of the 18th century (1782-1797) was briefly interrupted by

three years with average conditions in 1789-1791, followed by a 5-year drought since 1803-1809. One of the wettest periods in the northern sector of the Altiplano was recorded between 1818 and 1832, and the wettest year of the past 389 years was recorded for 1876, immediately followed by an extreme dry year in 1877. Decadal drought condition was recorded for the period 1926 to 1941, thereafter, another prolonged event with abundant rainfalls extended between 1942 and 1957. A marked negative trend in precipitation was recorded in the second half of the 20th century and the beginning of the 21st

century, culminating in a dry period from late 1970s decade to the present. The period from 1997 to 2013 is particularly dry with a mean precipitation value (89.8 %) at the lowest tail of the 17-year mean moving window precipitation distribution, that is a ~30% less precipitation than the historical mean calculated from 1625 to 2013 (120.3%) (Fig. 4a). Averaged precipitation for 1997-2013 ranks as the single driest 17-year period during the last 389-years (Fig. 4a).

**3.3. Temporal changes in dry/wet precipitation extremes**

The return-time analysis for the reconstruction is characterized by a non-stationarity behavior in the occurrence of dry extremes, with the highest rates recorded during the last decades of the 20th and beginning of 21st centuries with one extreme event recorded every ~9 years (Fig. 4b). The opposite situation occurred before 1650 with one extreme dry event recorded every ~33 years. The 18th and 19th centuries were relatively stable with a drought every ~20-27 years, while from



1940 to 2013, there was a steady increase in extreme dry events from ~17 to ~9 years (Fig. 4b). The return-time analysis of
abundant precipitation events varies between one pluvial event every 12-30 years (Fig. 4b). The highest rate of pluvials was
recorded since 1626 to 1680 with a wet event every ~12-15 years. Also, a high rate of extreme wet events occurrences (every
~15 years) were recorded during the first half of 19th century (Fig. 4b). The addition of extreme dry and pluvial events,
indicate that the highest occurrence rate of extreme events took place during the second half of 20th century and beginning
of 21st century with one extreme event every 6.7-7.5 years (Fig. 4b). This period is dominated by the occurrence of extreme
dry events. The second higher period of extreme event frequency is the 17th century with one event every 8-10 years
dominated by pluvial events (Fig. 4b).

**3.4. Spatial patterns and dominant spectral modes of the precipitation variability**

The spatial correlation analysis between the reconstructed and observed NDJ precipitation of the north South American
Altiplano and the gridded GPCP NDJ precipitation for the common period 1979-2013, showed consistent spatial patterns.
High correlation coefficients ($r > 0.4$, $p < 0.05$) were observed accross the South American Altiplano and projected to the
southern Central Andes. Positive correlations ($r > 0.3$, $p < 0.05$) were also recorded in the surrounding lowlands from the
Central Andes and in the equatorial sector of Brazil, while weaker positive correlations were observed within the region of
convection over the Amazon basin (~5º S–17.5º S / 72.5º W – 47.5º W; Vuille et al. 2012) (Fig. 5a,b).

The spectral density analysis for the reconstructed precipitation shows increased spectral power from inter-annual to
centennial scales (Fig. 5c). Spectral peaks are significant (95% confidence) at cycles centered on 129, 3.9, 3.4, 2.4 and 2.1
years (Fig. 5c), indicating that the reconstruction contains a strong high-frequency signal. The same analysis for the
instrumental precipitation detects high, but not significant, spectral power at inter-annual scales that ranges between 2.1 and
7.1 years spectral peaks (Fig. 5d), consistent with the high frequency spectral signal in the reconstruction.
The main secular (e), decadal (f) and interannual (g, h) oscillatory modes of the northern Altiplano precipitation
reconstruction were related to the following oscillatory cycles with the associate variance from the total precipitation
reconstruction: ~123 year with 17.3% variance (Fig. 5e), 9.2- 11.2 year with 20% variance (Fig. 5f), 3.1-3.9 year with 17%
variance (Fig. 5g) and 2.1-2.8 years with 17.6% variance (Fig. 5h). The changes in amplitude indicate a shift in the intensity
of the oscillatory waveforms. The secular wave is dominated by the occurrence of wet events around the 1820s and 1940s.
The decadal wave is most intense between 1625-1750, 1770-1840, 1870-1920. The oscillatory mode in the 3.4 years
waveform shows an increase in amplitude in the periods 1795-1845, 1875-1920, 1960-1970, 1990-2000, and a decrease in
amplitude during 1920-1960. The amplitude of the 2.4 years waveform was highly variable throughout the past four
centuries (Fig. 5h).
Regarding instrumental precipitation the inter-annual oscillatory modes of 3-4 years and 2-1 and 2.4 years explained 17.4%
and 30% of the total precipitation variance. Significant positive correlations were recorded at the main inter-annual
oscillatory modes of ~3.5 and ~2.4 cycles between the reconstruction and instrumental precipitation (Fig. 5g,h), showing the





high frequency variability shared by both precipitation records. More specifically, the inter-annual oscillatory modes of 3.1-3.9 year reconstructed precipitation and the 3-4 year instrumental precipitation correlated positively (r = 0.67; p < 0.01; Fig. 5g), as well as 2.1-2.8 year reconstructed precipitation and the 2.1-2.4 year instrumental precipitation (r = 0.51; P < 0.01; Fig. 5h).

**3.6. ENSO signals in the northern South American Altiplano precipitation**

For the following comparisons between tropical Pacific SST and precipitation for the northern Altiplano region, we used the reconstructed precipitation overlapping longer time with the SST time-series (1949-2013 and 1870-2013) than the shorter instrumental precipitation record (1970-2019). Figure 6 examine the influence of tropical Pacific SSTs variability over the north South American Altiplano. The spatial correlation pattern showed a significantly negative relationship between the
reconstructed precipitation and the SSTs from tropical central-east Pacific Ocean for the period 1949-2013. The highest correlations were found in the region south of the NIÑO 3 sector (~2º S – 13º S, 100º W – 150º W; Fig. 6a).

The time-series of reconstructed precipitation and the mean SSTs Oct-Sep anomalies averaged for the NINO 3 sector of the tropical Pacific showed significant negative correlations (r = -0.44, P < 0.001; Fig. 6b). Correlation coefficients calculated between the two time-series for time windows of 30 years lagged one year detected changes in the stability of the
relationship between the precipitation reconstruction and the SSTs_N3 throughout time (Fig. 6c). Strong correlations (r > -0.42) were recorded centered in the periods 1885-1900, 1940-1955, 1965-1975, while sharp decreases in correlations were found centered in the periods 1910-1935, 1978-2000 (Fig. 6c).

Both, reconstructed precipitation and SSTs_N3 recorded dominant oscillations modes at decadal (Fig. 6d) and inter-annual (Fig. 6e,f) frequencies explaining high percentage of their variability. A correlation analyses among the main dominant
oscillatory modes at decadal 10.2 year (Fig. 6d) and inter-annual frequencies 6.2 year (Fig. 6e) and 3.5 year (Fig. 6f) of the precipitation reconstruction and the SSTs_N3 with 12.7 year, 5.6 year and 3.6 year, respectively, showed in the three cases significant negative correlations. These results point out the occurrence of common waveforms and demonstrate the ENSO signal in the precipitation reconstruction from the northern South American Altiplano.

**3.7. Comparisons among hydroclimatic tree-ring based reconstructions in the Altiplano**

The NDJ precipitation reconstruction was compared with independent multi-century tree-ring based hydroclimate reconstructions from three different regions in the South American Altiplano (section 2.9 in Material and Methods). The correlations between records show significant relationships between our reconstruction and the three-hydroclimatic reconstructions generated using independent tree-ring records (Fig. 7). The highly significant correlations between the reconstructions reflect a strong co-variability in inter-annual and decadal hydroclimate fluctuations over all the South
American Altiplano from north to south during the past four centuries (Fig. 7). All four reconstructions show the occurrence of a period with inter-annual oscillations embedded in slightly wetter long-term conditions during the period ~1625-1750.



An extended period with dry conditions from ~1780 to1810 was followed by a very wet years between ~1820-1850. Dry conditions were documented in all four reconstructions during the mid-1920s and 1930s. However, the wet period recorded during the 1940s and 1950s by the northern Altiplano reconstruction presented here, which represent one of the two wettest

events in this new record, did not seem detected by the three other reconstructions located in southern localities (Fig. 7). Remarkably, the long-term interval with persistent droughts recorded during the last decades of the 20th century and beginning of 21st century is consistently observed in all the hydroclimatic records in the Altiplano, representing the period recording the most severe drought conditions over the entire centennial time span for each one of the hydroclimate reconstructions (Fig. 7).

**4. Discussion**

Here, we present the first tree-ring based reconstruction of precipitation variations over the last ~four centuries for the northern South American Altiplano region. This study provides new insight into the climatic variability of the tropical Andes based on the analysis of 389 years of rainfall in a region lacking long-term local meteorological records. To develop this reconstruction we used 132 tree-ring series from *P. tarapacana*, a species characterized by its high sensitivity to local and

regional hydroclimatic variations (Argollo et al., 2004; Morales et al., 2004; 2012; 2015; Christie et al., 2009; Solíz et al., 2009; Crispín-DeLaCruz et al., 2022). This newly developed precipitation reconstruction allow us to explore the evolution, periodicities and trends of rainfall variability during the last four centuries, as well as changes in the occurrence rate of extreme dry and wet years within a multi-century perspective for the northern region of the Altiplano.

The regression model used to develop the reconstruction explains 42.5% of the total variance of the late spring (November) to the mid summer (January) precipitation for the 1970-2010 calibration period. This value is comparatively lower than that previously reported for a tree-ring based precipitation reconstruction in the central-southern Altiplano ($R^2$ = 0.56; Morales et al., 2012). The lower sample size of trees used to generate the northern chronology in comparison with the southern-central reconstruction could have influenced the relatively lower correlation obtained and highlight the importance of fieldwork

expeditions to increase the number of centennial tree-ring chronologies in the northern sector of the Altiplano. Other factors such differences in hydroclimate conditions prevailing throughout the entire Altiplano and/or species sensitivity to precipitation should also be considered as potential causes for the observed difference in $R^2$. In the northern Altiplano, rainfall is almost double than recorded in the southern region (i.e. 300-400 mm vs. 100-200 mm, respectively; Crispín-DeLa Cruz et al., 2022), therefore precipitation would not be as limiting for tree-growth as in the more arid southern regions of the

Altiplano (Solíz et al., 2009; Rodríguez-Catón et al., 2021). While rainfall explains a high percentage of the total variance of radial growth, temperature variations also play an important role via evapotranspiration regulating stomatal aperture and carbon fixation for *P. tarapacana* growth as indicated by Rodríguez-Catón et al. (2021).



Prolonged dry periods in our reconstruction such as those recorded in 1782-1797, 1926-1941 and 1997-2013, highlight the
existence of decadal droughts in the northern South American Altiplano sector. Similarly, prolonged wet conditions were
reconstructed across this region during the periods 1818-1832 and 1938-1957. Most dry and wet periods recorded in our
reconstruction are consistent with those documented in a tree-ring-based precipitation reconstruction for the central and
southern Altiplano region (Fig. 7) for precipitation (Morales et. al., 2012) and a lake area size reconstruction for the
southwestern Altiplano (Morales et al., 2015), as well as a scPDSI reconstruction averaged for the entire Altiplano region
(Morales et al., 2020). This indicates the existence of a common general temporal pattern of hydroclimate variability across
the South American Altiplano region. However, our newly developed record have revealed some important regional
differences, such as the more pronounced humid conditions during the period 1938-1957 in the north than in the central and
southern sectors of the Altiplano (Fig. 7; Morales et al., 2012; 2015; 2020).

The first half of the 19th century recorded one of the wettest periods over the past 389 years, consistent with the wet
conditions identified in the other three tree-ring-based hydroclimatic reconstructions from different regions of the Altiplano.
Similarly, other lower resolution environmental proxies such as the speleothem record from the eastern Bolivian Andes
caves (Apaéstegui et al., 2018), the sediment core from the peat accumulation wetland at Cerro Tuzgle (Schittek et al.,
2016) and environmental records derived from deposits of rodent plant debris in the Andean foothills of the northern
Atacama Desert in Chile (Mujica et al., 2015), highlight the occurrence of this extreme pluvial period during the 19th
century. This pluvial maximum coincides with a significant population increase of the Aymara people from the Tarapacá
region in the Chilean Altiplano (Lima et al., 2016), which would have been triggered by increased productivity in their
agropastoral system as a result of favorable wet conditions. The wettest year recorded in our reconstruction is 1876, which is
associated with the occurrence of a prolonged cool phase of the central tropical Pacific during 1870–1876 period (La Niña
conditions), being the year 1876 the coldest SST record (Singh et al., 2018). This cool-La Niña phase conditions reversed to
a warm SSTs during the strong 1876/1877 El Niño event (Singh et al., 2018), which is registered in our precipitation
reconstruction as an extreme dry year in 1877.

Since the mid-1970s, precipitation in the northern Altiplano recorded a sustained negative trend, with the 1997-2013 period
stands out as exceptional, being the driest period of the last 389 years. For instance, the reconstructed south-central
precipitation, the southwestern lake size, and the average scPDSI for the Altiplano also highlight the occurrence of arid
conditions during this late 20th-beginning 21st centuries period (Fig. 7). These results are consistent with the rapid retreat of
glaciers across the tropical Andes during the second half of the 20th century (Ramírez et al., 2001; Francou et al., 2003;
Vuille et al., 2008; Jomelli et al., 2009). On the other hand, this abrupt change towards more arid conditions was also
recorded since the 1970s onwards in the Quelccaya ice core (Thompson et al., 2006), the Pumacocha sediment (Bird et al.,
2011) and the Paleo Hydrodynamics Data Assimilation (PHYDA; Steiger et al., 2018) for the central-south Altiplano region
(20º-23º S, 66º-68.5º W; Fig. S2). It is important to place this increase in aridity conditions since the late 20th century –
beginning 21st century observed in our northern record in the long-term context and the great spatial coherence and



synchrony shown by all the other proxies records across the southern tropical Andes (Fig. 7 and Fig. S2), suggesting large-scale common atmospheric and ocean forcings over this Andean region.

The BT spectral density analyses highlight that the rainfall reconstruction shares strong inter-annual periodicities with the instrumental precipitation index. At an inter-annual frequency band, the dominant oscillatory modes found for the northern Altiplano precipitation are consistent with modes described in hydroclimatic reconstructions developed for the central and southern Altiplano (Morales et al., 2012; 2015), while sub-decadal and multi-decadal periodicities were important but not significant such as in the previous two reconstructions. The main waveforms of variability that compound the northern
Altiplano NDJ precipitation reconstruction showed non-stationary cycles in precipitation during the last 389 years. The 17% of the rainfall variance is associated with the oscillatory mode of 3.1-3.9 years cycle that exhibits some peculiar cyclic properties, showing the highest amplitude cycles over the late 1700s through 1800s and some periods of 1900s. In contrast, the amplitude variability of the waveform decreased in the period 1920-1960, which is consistent with the low amplitude variability of the central and southern Altiplano hydroclimatic reconstructions (Morales et al., 2012; 2015), which has been
associated with low ENSO activity reported during the same period (Aceituno and Montecinos, 1993).

Our rainfall reconstruction showed consistent spatial patterns with rainfall across the entire Altiplano and southern Central Andes, but also with the equatorial region of Brazil and the core sector of maximum monsoon convection activity, that reflected the SASM influences over the north Altiplano precipitation. In the same way a clear influence of tropical Pacific
SSTs in the reconstructed precipitation was detected, in particular from the N3 sector, while the rainfall reconstructions developed in the central and southern Altiplano show a closer link mainly with SSTs from the N3.4 sector (Morales et al., 2012; 2015). This could reflect a latitudinal gradient of rainfall along the Altiplano associated to the spatial pattern variability of SSTs in the tropical Pacific. Both, the reconstructed rainfall and the SSTs_N3 shared spectral properties at decadal, sub-decadal and inter-annual variability that also demonstrating the ENSO signal imprints in precipitation at
different time scales. A high consistence was observed in the amplitude changes in the main oscillatory modes of decadal and inter-annual variability between reconstructed precipitation and SST_N3, reflecting in these bands the instability of ENSO's influence on local precipitation.

This new precipitation reconstruction provides also a long-term context for our present understanding of hydroclimate
extremes in this region and identifies an increase in the occurrence rate of extreme droughts since the mid-20th century. Climate change is already affecting extreme events across the globe (IPCC, 2021) and the north Altiplano is not an exception. According to our reconstruction, since 1990s the occurrence of one extreme dry year every nine-ten years, an occurrence rate two and three times higher than the rates reported for 19th, 18th and 17th centuries. The instrumental regional precipitation index developed in this study show a mean condition value (100 % w.r.t. 1982-2013) that represents



20% less precipitation than the reconstructed mean (120 % w.r.t. 1625-2013). Therefore, the period corresponding to the instrumental data represent a dry condition in relation to the entire reconstructed 389 years record.

The multi-decadal negative trend observed in our precipitation reconstruction is consistent with other hydroclimatic reconstructions based on tree rings and other paleoclimatic proxies for the region in a long-term perspective. Nonetheless, there are some inconsistencies with studies using short instrumental precipitation records. Segura et al. (2020) based on

instrumental-satellite precipitation data for the southern region of the tropical Andes (12º-20º S; 60º-80º W), evaluate the common pattern of summer rainfall variation for this region during the period 1982-2018, identifying a positive trend specially based in positive anomalies after 2010 that would be influenced by upward motion over the western Amazon. This study included our study area but the much larger domain used in the Segura's study may not fully represent the local rainfall variations from our study area. This difference between trends can also be due to distinct time span of the

instrumental records used. Because trends are timescale dependent comparing instrumental precipitation over the recent decades with long-term paleoclimate records provides a more comprehensive perspective than assessing trends for only the past few decades in isolation.

Droughts are of particular relevance in climate variability for this semi-arid region of the Andes. Under a global warming

context, the Altiplano's water resources are fundamental for biodiversity conservation and socioeconomic activities. The projected increase in evapotranspiration via global warming, together with a wide range of variability among the precipitation models projected for the 21st century, may very likely lead to growing demand for water in this region already under pressure for water scarcity. A better understanding of the future of Altiplano's water resources should be listed as priority for stakeholders and decision-makers to avoid social conflicts at both the local and regional levels. Under this

complex political, social and environmental scenario, it is highly relevant to anticipate the possible occurrence of these hydroclimatic changes to provide information for improving water resource policies in order to plan and implement adaptive strategies to reduce these vulnerabilities in the face of future water shortages.

**4. Conclusions**

In this study we developed the first tree-ring reconstruction of late spring-mid summer precipitation for the northern South American Altiplano. Our reconstruction covers the last 389 years and extends dendroclimatological studies northward in the tropical Andes, filling a gap in paleoclimatic information at low latitudes.

Our study provides an overview of the hydroclimate of the northern South American Altiplano through the identification of long-term wet or dry periods and the temporal evolution of annual precipitation during the last ~four centuries. In addition,

the secular and inter-annual dominant oscillatory modes in precipitation were identified.



Extreme dry years have been more frequent during the second half of the 20th century in the northern Altiplano within the context of the last 389 years. This is in agreement with the reported increase in the occurrence of extreme dry event years in other regions across the Andes.

Persistent periods of drought/wet conditions over the past 389 years are highly consistent with evidences provided by
independent paleoclimatic records available for the Altiplano. Our study revealed a strong common temporal and spatial synchrony of drought conditions since 1980s recorded by different tree-ring based hydroclimate reconstructions across the Altiplano region and by other paleo-proxies of hydroclimate variability from the southern tropical Andes.

Our 389-year long rainfall perspective allows us to evaluate the 1970-2019 period covered by the instrumental records in this region of the Andes in a multi-century context. A shift in dry conditions in our precipitation reconstruction since the mid
1970s suggests that the 49-year interval of instrumental records coincides with this multi-decadal long dry period in the northern Altiplano. This indicates that the mean condition of this recent period has lower precipitation amounts than the historical mean and thus, it is not fully representative of the natural envelope of the precipitation regime in this Andean region observed during the last centuries.

**Author contribution**. M.S.M., E.J.R.-R., D.A.C., L.A.-H., R.V. and M.E.F. designed research; M.S.M., E.J.R.-R. and D.A.C. collected samples; M.S.M. performed research; D.B.C.C., C.A., D.A.C. and E.J.R.R. provided tree ring measures and interpretation; M.S.M., D.B.C.C., A.G., R.LL.M., G.T.-O, J.S.-F. and E.R.-R .analyzed data; M.S.M. wrote the paper; D.A.C., R.V., M.E.F. and L.A.-H. reviewed and edited versions of the paper.

**Competing interest**: The authors declare no competing interest.

**Data availability:** Instrumental and reconstructed precipitation data set, as well all the data used to develop figures 2, 3 and 4 will be deposit at the https://www.ianigla.mendoza-conicet.gob.ar/portal1/ upon the manuscript have been accepted for publication.

**Acknowledgements.** We acknowledge the Dirección General de Aguas and Dirección Meteorológica de Chile in Chile, and Servicio Nacional de Meterología e Hidrología in Bolivia and Perú for providing quality-controlled daily and monthly climate data, which were essential for developing the reconstruction model. This study was supported by the Fondo Nacional de Desarrollo Científico, Tecnológico y de Innovación Tecnológica, Perú (FONDECYT-BM-INC.INV 039-2019). L.A.-H.
and M.S.M. were supported by the US National Science Foundation (NSF) AGS-1702789, AGS-1903687 and OISE-1743738, USA, and by The THEMES project funded by the BNP Paribas Foundation in the frame of its 'Climate Initiative' program. M.S.M., M.E.F. and R.V. were supported by the Agencia Nacional de Promoción Científica y Tecnológica, Argentina (PICT 2013-1880), Consejo Nacional de Investigaciones Científicas y Tecnológicas (PIP 11220130100584)



projects. D.A.C. was supported by FONDECYT 1201411, FONDAP 15110009, ANID/BASAL FB210018. M.E.F was
partially supported by ANPCyT-PICT-2019-01336.

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



Tables.

Table 1. Geographical locations of *P. tarapacana* sampling sites in the Altiplano of Peru (Chiluyo, CHI; Paucarani, PAU), and Chile (Suriplaza, SUR), statistics for the individual and regional chronologies, and correlation matrix for the common 1880-2007 period (all r values $P<0.05$).

| Chronology | CHI | PAU | SUR | Regional |
|---|---|---|---|---|
| Period | 1602-2015 | 1787-2015 | 1615-2007 | 1602-2015 |
| Latitude (S) | 17º 24' | 17º 34' | 17º 51' | |
| Longitude (W) | 69º 39' | 69º 45' | 69º 28' | |
| Altitude (msnm) | 4657 | 4696 | 4800 | |
| Nº series | 52 | 22 | 58 | 132 |
| Mean series corr. | 0.60 | 0.61 | 0.57 | 0.56 |
| Rbar | 0.40 | 0.44 | 0.40 | 0.36 |
| EPS | 0.92 | 0.92 | 0.91 | 0.94 |
| CHI | - | | | r = 0.87 |
| PAU | r = 0.65 | - | | r = 0.78 |
| SUR | r = 0.59 | r = 0.67 | - | r = 0.88 |






**Table 2.** Geographical locations of the meteorological stations in the Altiplano used in this study to compose the regional precipitation and temperature indexes for the northern Altiplano.

| Code | Name | Lat (S) | Long (W) | Country | Altitude (m a.s.l.) | Parameter | Period |
|------|------|---------|----------|---------|---------------------|-----------|--------|
| Pam | Pampahuta | 15.5 | 70.68 | Peru | 4400 | Temp | 1962-2013 |
| Lam | Lampa | 15.36 | 70.37 | Peru | 3892 | Temp | 1964-2013 |
| Pun | Puno | 15.83 | 70.01 | Peru | 3812 | Temp | 1964-2012 |
| Ila | Ilave | 16.08 | 69.64 | Peru | 3871 | Temp | 1984-2013 |
| Des | Desaguadero | 16.56 | 69.04 | Peru | 3808 | Temp | 1956-2013 |
| Alt | El alto | 16.51 | 68.20 | Bolivia | 4034 | Temp | 1943-2015 |
| Maz | Mazocruz | 16.74 | 69.72 | Peru | 4003 | Prec | 1964-2013 |
| Col | Collana | 16.90 | 68.28 | Bolivia | 4500 | Temp | 1973-2015 |
| Cap | Capazo | 17.19 | 69.74 | Peru | 4530 | Temp | 1964-2013 |
| SanM | Santiago de Machala | 17.07 | 69.20 | Bolivia | 3883 | Temp Prec | 1981-2015 1979-2019 |
| Vis | Visviri | 17.6 | 69.50 | Chile | 4080 | Prec | 1968-2019 |
| ViI | Villa Industrial | 17.78 | 69.71 | Chile | 4080 | Prec | 1976-2018 |
| Alc | Alcerreca | 18.00 | 69.65 | Chile | 3990 | Prec | 1971-2018 |
| Put | Putre | 18.20 | 69.56 | Chile | 3545 | Prec | 1970-2019 |
| Chua | Chungara Ajata | 18.23 | 69.18 | Chile | 4585 | Prec | 1982-2019 |

Source: Peru: SENAMHI; Bolivia: SENAMHI; Chile: DMC and DGA.




**Figure Captions**

Figure 1. Study area. (a) The northern Altiplano region showing the geographic locations of the three *P. tarapacana* ring-
width chronologies (red circles) and the 15 weather stations (blue triangles) used in this study to create the regional
precipitation and temperature indexes. Basemap source: GADM 2022 and data version 4.0.4. were downloaded
from https://gadm.org/data.html. (b-d) Study sites: (b) Chiluyo, (c) Paucarani and (d) Suriplaza.

Figure 2. Comparison between the inter-annual variations of the tree-ring chronology with, (a) the regional precipitation
index for the period 1970-2014 and (b) the regional temperature index for the period 1963-2014 for the north South
American Altiplano. The temperature is shown inverse to facilitate the comparison between records (right axis). The
correlation coefficients and the slope trend and significance are indicated in each panel.

Figure 3. Precipitation variations in the northern sector of the Altiplano during the last four centuries. (a) Observed
(instrumental) and estimated (tree rings) variations of late spring-summer precipitation (November-January) in the northern
part of the Altiplano (precipitation anomalies are expressed in percentages with respect to the calibration period 1970-2010).
Calibration and verification statistics: explained variance ($R^2$) during the calibration period, Pearson's correlation coefficient
(r) between observed and reconstructed values, regression F-value and reduction of error (RE). (b) The residuals of the
regression (red line) with the slope of the trend (black line). The Durbin-Watson (DW) statistic and the trend slope are
indicated. (c) Tree-ring based reconstruction of late spring-summer precipitation in the northern Altiplano region for the
period 1625-2013 (precipitation anomalies expressed in percentages (%) with respect to the average instrumental
precipitation for the common period 1982-2013. To emphasize the low-frequency variations, a 35-year cubic smoothing
spline reducing 50% of the reconstruction variance (red line) is also shown. The gray band represents the RMSE. Vertical
red (blue) bars in the bottom of the panel indicate the extreme dry/pluvial events calculated for the 5th and 95th percentile
thresholds, respectively.

Figure 4. 1997-2013 drought and extreme precipitation occurrence rate events. (a) Average reconstructed precipitation for
the period 1997-2013 compared to 17-year average moving window precipitation anomalies distribution (%) for the
reconstructed period 1625-2013. (b) Occurrence rate of extreme dry (orange) and wet (blue) precipitation events and the sum
of both (dry+wet; gray) for the period 1625-2013. Extreme dry/wet events were calculated for the 5th and 95th percentile
thresholds of the precipitation reconstruction, respectively. A smoothed bandwidth of 40 years was applied using a time-
dependent Kernel method. The shaded areas represent 95% confidence intervals based on 1000 bootstrap simulations.

Figure 5. Spatial patterns and spectral properties of precipitation. Spatial correlation field between the GPCP 2.5° x 2.5°
gridded NDJ precipitation and, the northern Altiplano (a) reconstructed and, (b) instrumental NDJ precipitation for the





period 1979-2013. (c) Blackman-Tukey power spectral analysis of the reconstructed (1625-2013) and (d) instrumental (1970-2019) precipitation from the northern sector of the South American Altiplano. Black numbers indicate periods with high spectral power but not statistically significant (95% cl), while orange numbers represent significant peaks (95% cl). (e-h) Dominant modes of secular, decadal and inter-annual variability extracted by Singular Spectral Analysis (SSA) from the
reconstructed and instrumental precipitation over the period 1625-2013 and 1970-2019, respectively. The frequencies of the SSA waveforms for the reconstructed (instrumental) precipitation are indicated in black (blue) number of years with the corresponding explained variance in percentages (%).

Figure 6. ENSO signals in the northern South American Altiplano. (a) Spatial correlation patterns over the period 1949-2013
between the reconstructed November-January (NDJ) precipitation for the northern Altiplano (black square) and the gridded 2.5º x 2.5º October-September (O-S) sea surface temperatures (SSTs). (b) Inter-annual relationship between the reconstructed NDJ precipitation and O-S sea surface temperatures (SSTs) averaged for the NINO 3 region (SSTs_N3; white rectangle in panel a) for the period 1870-2013. (c) Moving 30-year correlation between the reconstructed NDJ precipitation and the SSTs_N3 for the period 1870-2013. (d-f) Comparisons between the waveforms of the precipitation reconstruction
(black line) and the SSTs_N3 (red line) extracted by Singular Spectrum Analysis (SSA) over the 1870-2013 period. The periodicities are indicated in each panel and the percentages of variance explained by each frequency are indicated in parentheses. The correlation coefficients between the two series are shown at each panel.

Figure 7. Comparison among four-hydroclimate tree-ring based reconstructions from the South American Altiplano. From
top to bottom: (a) precipitation from the northern sector of the Altiplano (this study), (b) lake area variations from the southwestern Altiplano, (c) scPDSI from the entire Altiplano (17º-23º S; 66º-70º W), and (d) precipitation from the central and south Altiplano. Shaded background represents long-term drought (yellow) and wet periods (light blue) coincidences among the four reconstructions. Correlation coefficients (r and p values) between the northern Altiplano precipitation reconstruction and each hydroclimatic reconstruction are indicated with black numbers in their corresponding panels for the
common period (1625-2008).





Figure 1


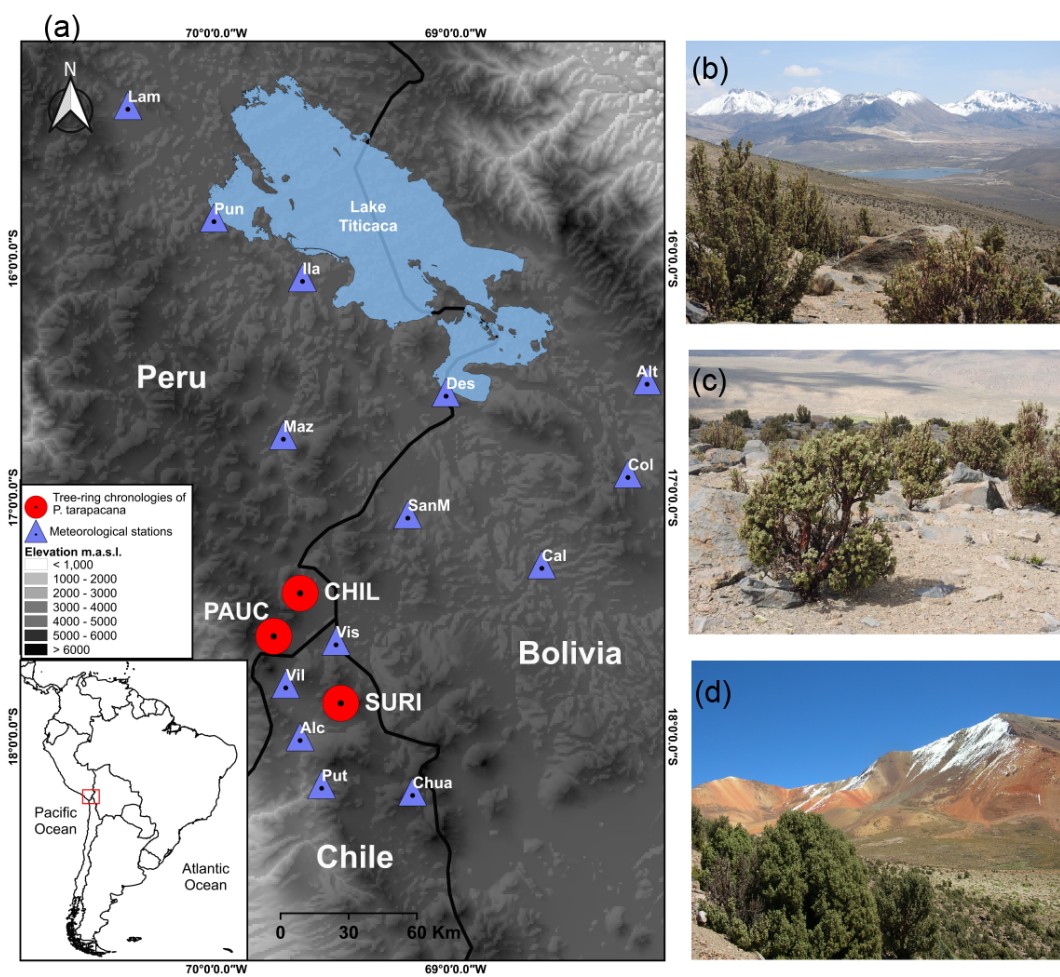



Figure 2


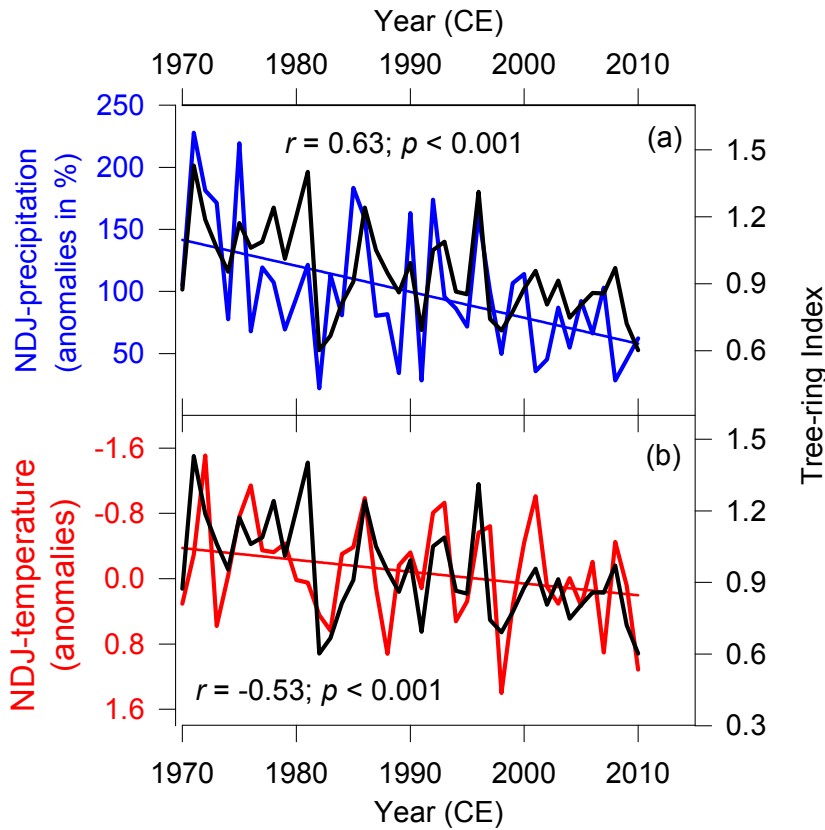



Figure 3


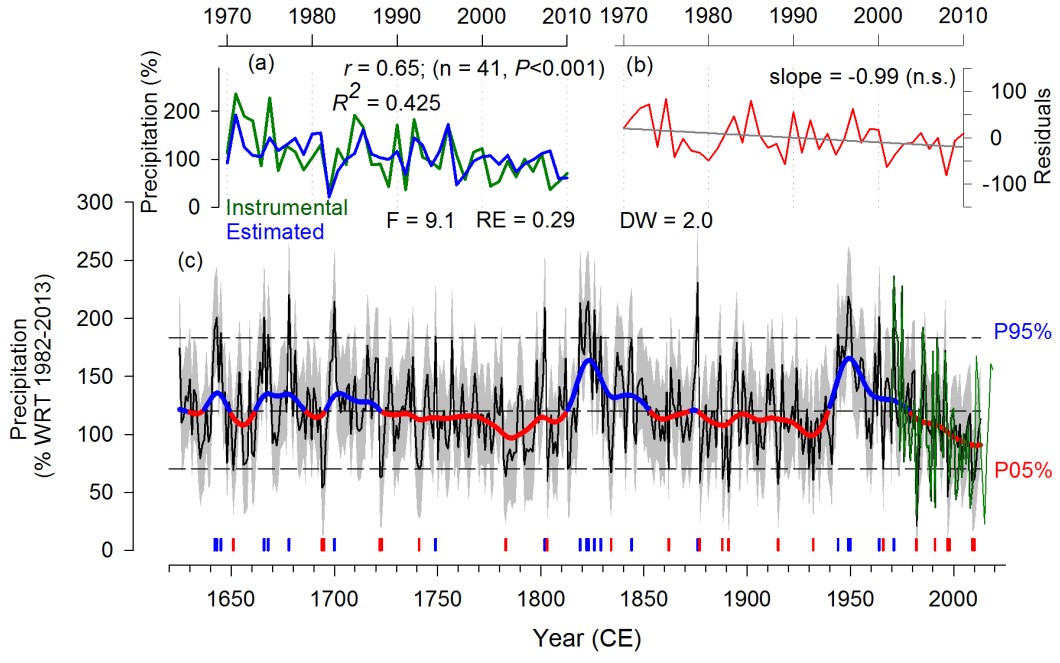






Figure 4

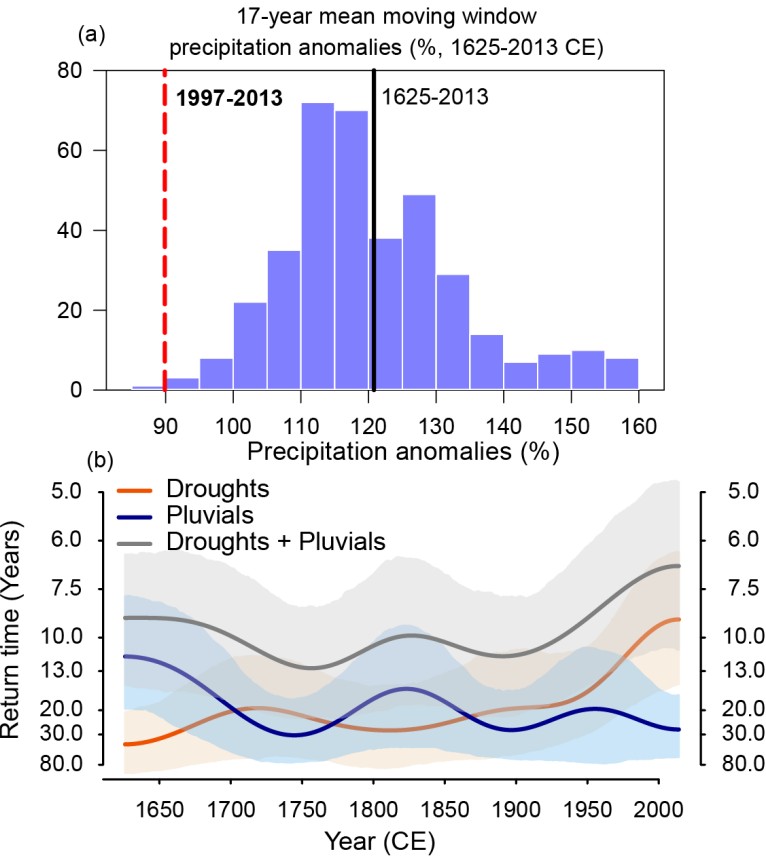



Figure 5.

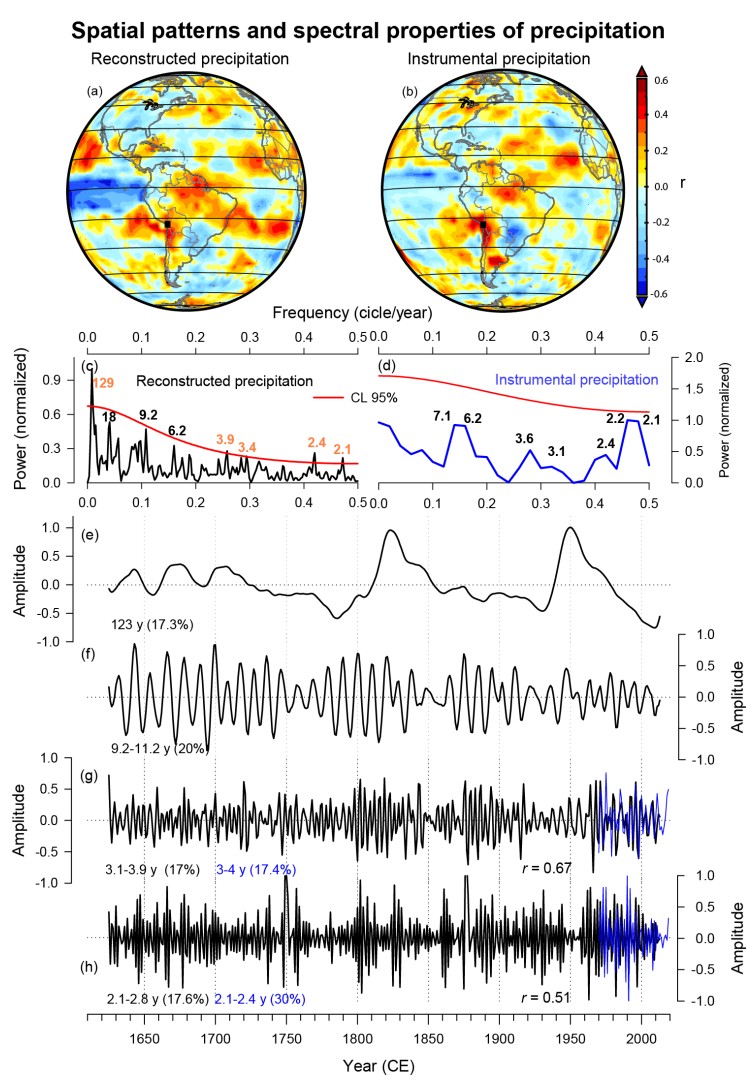





Figure 6.

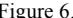

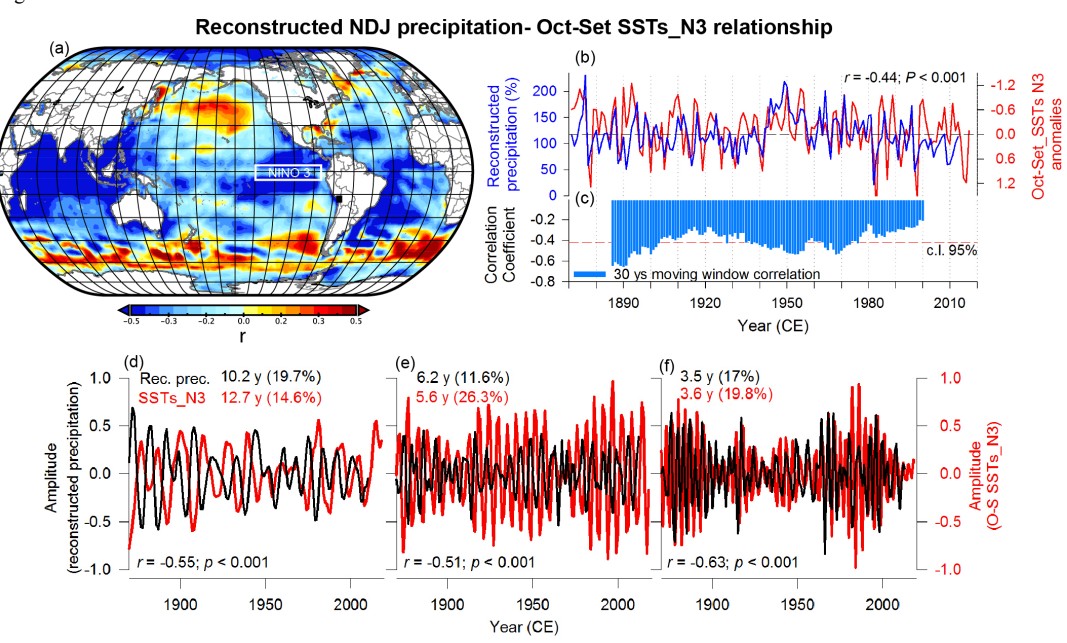



Figure 7.

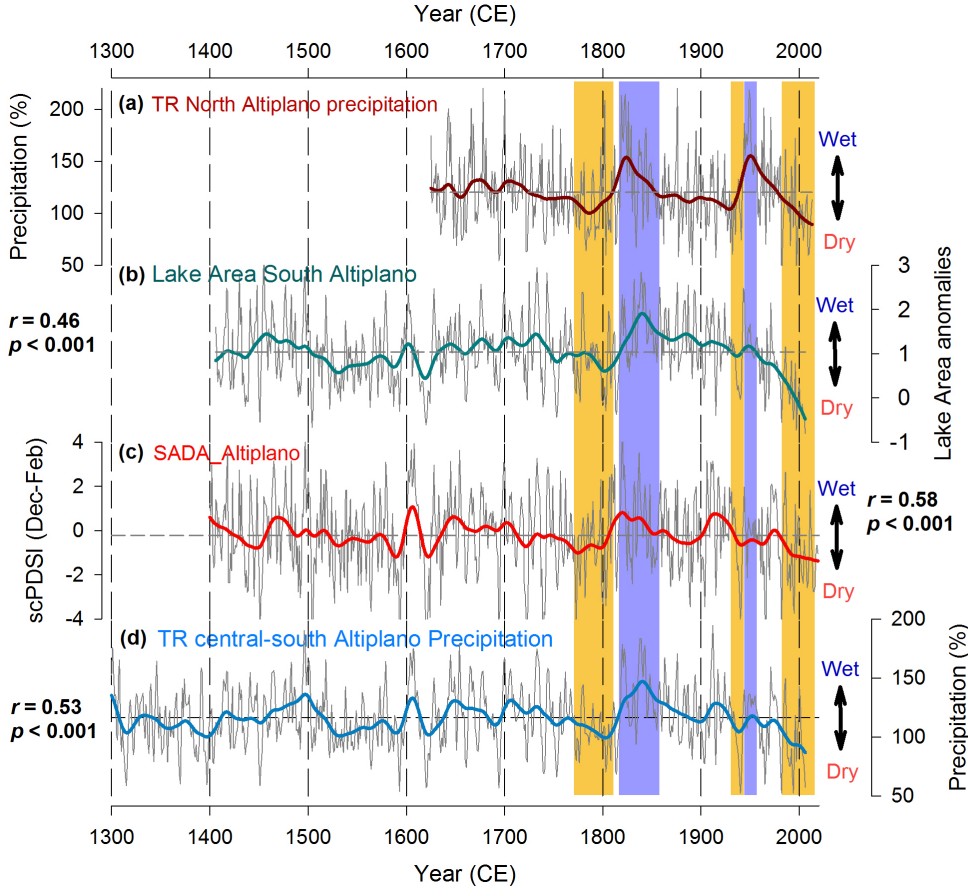
