# Peer review of "Drought increased since the mid-20th century in the northern South American Altiplano revealed by a 389-year precipitation record"

_Climate of the Past, 2022_

## Referee Comment (RC2)

[Figure]

[Figure]

Figure 1

[Figure]

[Figure]

Figure 2

[Figure]

[Figure]

Figure 3

[Figure]

[Figure]

[Figure]

Figure 4

[Figure]

[Figure]

[Figure]

Figure 5.

[Figure]

[Figure]

[Figure]

Figure 6.

[Figure]

[Figure]

Figure 7.

[Figure]

[referee-annotated manuscript omitted]

---

## Author Comment (AC1)

Dear editor Keely Mills,

Please, find below our responses to the reviewer comments on our submission to *Climate of the Past*, entitled "Drought increased since the mid-20[th] century in the northern South American Altiplano revealed by a 389-year precipitation record".
We have prepared a new version of the manuscript that addresses most of the comments from the reviewer. In preparing this new version, we have included the two figures that were in the SI to the main text. Figure S1 is now part of Figure 2, while Figure S2 is now Figure 8.
Reviewer comments are *italicized* and in quotes, our responses are in blue, original text are in black and any substantive changes to the manuscript text are in red.

We thank the reviewer for their thoughtful critiques and his time and consideration.

Kind Regards,
Mariano Morales (corresponding author)

**Reviewer #1**

General comments

*"I understand that your main point is to defend the idea of an increased dryness conditions in your study area, based on a tree-ring precipitation reconstruction, however I'm not completely sold on this idea, and you show a brief discussion about it at the end of your discussion in which you mention there are studies (one at least), that show an opposite trend for the same time period, over your study area.*
*I think one of the key messages would be why would we value the tree-ring based reconstruction of precipitation you are offering over the instrumental data that are currently available. You mention the other paleoclimate proxies, however they have problems like trends that are difficult to remove sometimes."*

We consider that understanding the variability (intensity and frequency)  of extreme hydroclimatic events in the Altiplano is a current topic of debate. The particular geographical location and topographic complexity of the Altiplano creates contrasting climatic conditions throughout the region. This enormous environmental complexity induces contrasting climatic trends in relatively close sites, thus limiting a solid understanding of the Altiplano's hydroclimatic variability. The low geographic density, the short-term nature of the records, frequently accompanied by the poor quality of the meteorological measurements, prevent us from capturing the complex hydroclimatic variability in the Altiplano, providing in consequence, a limited view of the mean fields and extremes of the natural hydroclimatic variability.
Although numerical and statistical approaches have improved regional future projections, the sign of future trends in water availability still remains difficult to estimate. Therefore, we believe that a long-term perspective of natural hydroclimatic variability based on climate proxies is needed to complement and contextualize present instrumental data. The driest conditions recorded in our reconstruction during the last four-five decades are also presented in independent hydroclimatic reconstructions developed for the Andean domain. Consistent with our results, the sharp shrinkage -and in some cases the disappearance- of tropical glaciers reflects this increase in aridity conditions since the middle of the 20th century.

We have included this point in the Discussion section (line 402 to 413) and added an additional paragraph in which we put forward different hypotheses about the causes of aridization in the Central Andes (line 414-425).

*Abstract:*
*"Lines 35-40: I missed a little explanation of what would be the influences of ENSO in the precipitation in your study area."*

We have added an additional paragraph to the "Abstract" to expand upon this point.

The rainfall reconstruction provides also valuable information about the ENSO influences in the northern Altiplano precipitation. Past drier conditions in our study region were associated with ENSO warm events. The spectral properties of the rainfall reconstruction showed strong imprints of ENSO variability at decadal, sub-decadal and inter-annual time-scale, in particular from the Pacific Niño 3 sector.

*"In Methods you mention in "Climate data collection and analysis" that you used stations with less than 10% missing data. Why did you make the average of the 7 stations? Did you analyze each station against your chronologies? Or against your regional chronology?"*

To be clearer we have reworded the text describing the steps to create the regional precipitation index and added an additional paragraph to the "Climate data collection and analysis" section.

In semi-arid environments with complex topography, such as the central Andes, precipitation has considerable spatial variability. Therefore, to minimize the influence of any single station record and to highlight the common pattern of spatial and temporal variation in the study region, we developed a regional precipitation index. In the northern Altiplano there are few instrumental records prior to 1970 and they are not regularly distributed. Monthly precipitation records were collected from seven meteorological stations located above 3200 m a.s.l. in the northern Altiplano between 16° and 19° S. Only station records with less than 10% of missing data were considered (Table 2). Missing data from individual stations were estimated using linear regressions (Ramos-Calzado et al., 2008). Correlation coefficients between the standard regional chronology of *P. tarapacana* and precipitation monthly variations from each of the seven meteorological stations were used to define the seasonal precipitation period best related to radial growth. Precipitation from November to January (NDJ, late spring – mid-summer) is the most strongly associated with tree growth. Inter-annual time series of the total NDJ precipitation were created for each of the seven available stations. Finally, the precipitation data from each station was normalized (z-score; Jones and Hulme, 1996) and averaged to obtain a regional index of precipitation from the northern South American Altiplano for the period 1970-2019. To assess the partial influence of temperature on tree growth we compiled monthly temperature data from nine high-altitude (>3200 m a.s.l.) meteorological stations from the Altiplano (Table 2). Based on these data, we computed a regional mean NDJ temperature index for the period 1963-2014 across the study region, following the same procedure described for developing the regional precipitation index.

*Did you try gridded data too?*

Yes, we did. However the strongest precipitation - ring width relationship was observed with instrumental records from weather stations and not with gridded products. Gridded precipitation data are spatially interpolated from instrumental data from local weather stations, but also often include remote stations, so the rainfall signal in tree rings could be relatively lower. Notwithstanding, the GPCP 2.5º x 2.5º gridded product was used to determine the spatial representativeness of our reconstruction and the instrumental regional precipitation record (please, see Figure 5a and b).

*"In line 197, did you try monthly correlations too? I'm confused, did you calculate the average of the entire year?"*

We have added a sentence that explains more clearly the process of selecting the SST months best correlated with the rainfall reconstruction. From "ENSO signals in the northern Altiplano precipitation", first paragraph:

To assess the influence of the tropical Pacific sea surface temperatures (SSTs) on the reconstructed NDJ precipitation variability over the northern Altiplano, we computed spatial correlations maps between the reconstructed NDJ precipitation and the SSTs from the gridded NCEP reanalysis global database (2.5º x 2.5º grid point resolution; Kistler et al., 2001) for the period 1949-2013. Here we test the relationships between the reconstruction and the SSTs on a monthly basis and for different sets of SSTs months: NDJ, DJF, NDJFM, Jul-Jun, Oct-Set. We found the strongest spatial correlation for the Oct-Set SST period. Since the SST shows a strong intra-annual persistence, it is expected to find a SST signal covering the whole year and not exclusively during the NDJ months corresponding to the precipitation reconstruction.

*"In line 221, is this your best correlation result? What period is this correlation?"*

Yes, this is the best correlation result for both precipitation and temperature for the common period 1970-2010, indicated in line 250-251.

*"In line 253 you say "stable", however I think you are comparing this ~ 20 years frequency with the increased frequency you observe in the 20-21th centuries, is that correct? Stable to me sounds like nothing was happening, however it was, but at a lower frequency. I suggest you change the text in this paragraph, there is no such a thing as stable in climate, because it is constantly changing."*

In this paragraph we use the word "stable" to refer to the fact that the recurrence time of extreme events remains relatively low and constant during the 18th and 19th centuries. In accordance with the reviewer's comment, we have eliminated "stable" and reworded the paragraph.

The 18th and 19th centuries showed a relatively low and constant recurrence rate of extreme dry events with a drought every ~20-27 years, while from 1940 to 2013, there was a steady increase in extreme dry events from ~17 to ~9 years (Fig. 4b).

*"In lines 345-347 I ask myself, what are the odds of your chronologies showing decreased growth due to increased temperature and not exactly related to decrease in ppt?"*

In lines 364-378 we state that the rainfall reconstruction model for the northern Altiplano region explained a lower percentage of total variance with respect to other reconstructions developed in the Altiplano and we put forward some alternative hypotheses about the possible causes of this difference in the R2 values.

Our model explains 42.5% of the total rainfall variations, a relatively lower percentage than expected by other tree-ring based hydroclimate reconstructions for the Altiplano. Therefore, there are additional factors related to tree growth and meteorological precipitation information that could be affecting the skill of our model, such as:

1- The low number, quality, spatial coverage and brevity of the instrumental records used to develop the reconstruction model.

2- For instance, NDJ precipitation is the best-correlated period with tree-growth, but other months also influence growth in a less proportion.

3- The lower number of tree samples used to generate the northern Altiplano chronology compared to the south-central reconstruction.

4- Differences in hydroclimate conditions prevailing throughout the entire Altiplano and/or species sensitivity to precipitation should also be considered as potential causes for the observed difference in R2

5- There are others climatic variables such as temperature and wind that contributes to increase evapotranspiration from plants and soil evaporation demand, generating more severe dry conditions. Trees integrate the responses to these biotic and abiotic factors in their growth. However, using dendrochronological methods we can determine which of these factors is most closely related to interannual growth variations. In the Altiplano, rainfall plays a major role in tree growth, as we have already documented in previous work (Morales et al. 2004; 2012; Solíz et al. 2009; Rodriguez-Catón et al. 2021; Crispín De La Cruz et al. 2022). In the Altiplano, precipitation and temperature co-vary inversely: radiation is higher on clear days without convection inducing an increase in temperature, which in turn leads to higher evapotranspiration and a decrease in water availability affecting tree growth.

Finally, the comparison with other independent paleoclimatic records of precipitation in the Altiplano is important to reduce uncertainties. In Figure 7, we show the good relationships between the precipitation reconstructions developed for the Central and Southern Altiplano with our reconstruction for the northern sector.

We have reworded the second paragraph of the discussion, mentioned other factors such as the quality of meteorological records that could contribute to the recorded R2 differences among the different reconstructions across the Altiplano. In addition, we have reduced the emphasis on secondary factors associated with tree growth that may affect the strength of relationship between growth and precipitation.

The regression model used to develop the reconstruction explains 42.5% of the total variance of the late spring (November) to the mid summer (January) precipitation for the 1970-2010 calibration period. This value is comparatively lower than that previously reported for a tree-ring based precipitation reconstruction in the central-southern Altiplano (R2 = 0.56; Morales et al., 2012). The lower sample size of trees used to generate the northern chronology in comparison with the southern-central chronologies could have influenced the relatively lower correlation obtained and highlight the importance of fieldwork expeditions to increase the number of centennial tree-ring chronologies in the northern sector of the Altiplano. The poor quality, low spatial coverage and brevity of the instrumental records used to develop the

reconstruction model, could also affect the relationships with tree-ring records. Other factors such differences in hydroclimate conditions prevailing throughout the entire Altiplano and/or species sensitivity to precipitation should also be considered as potential causes for the observed difference in R2. In the northern Altiplano, rainfall is almost double than that recorded in the southern region (i.e. 300-400 mm vs. 100-200 mm, respectively; Crispín-DeLa Cruz et al., 2022). Therefore, precipitation at our study site would not be as limiting for tree-growth as in the more arid southern regions of the Altiplano (Solíz et al., 2009; Rodríguez-Catón et al., 2021). While rainfall explains a high percentage of the total variance of radial growth, temperature variations could also play a secondary role via evapotranspiration regulating stomatal aperture and carbon fixation for *P. tarapacana* growth as indicated by Rodríguez-Catón et al. (2021).

*"In line 370 you mention 1876/1877, and according to your dated chronology, your 1876 ring is the one that started in November of 1876 and extended to 1877. That would be already when the El Nino happened, and yet you say your 1876 shows as the wettest year."*

Thank you very much for pointing out this dating error. The Niño event occurred during calendar years 1877/78 (Singh et al. 2018) and not in 1876/77 as stated in the original text. Therefore, the occurrence of the 1877/78 Niño event coincides with the occurrence of an extreme drought event recorded in the 1877 year of our reconstruction. The year 1877 of our reconstruction represents the months corresponding to November and December 1877 and January 1878.

We have changed the date in the manuscript, line 400.

*"In line 385 I would suggest you write what BT stands for, I know you already showed in methods, but it's a long way for the readers to go back and find the abbreviation."*

We added the complete name for the Blackman-Tukey analysis.

*"In paragraph 409-416: What is the main source of precipitation in the study area? Is it the Pacific Ocean or the glaciers? Is this drying trend related to losses in ice cover, increased temperature, or both, and are you suggesting increased SST's activity?"*

In line 55 of the Introduction we briefly describe the main source of precipitation in the Altiplano. However, we considered pertinent the reviewer's suggestion and decided to incorporate a new section in Methods describing the main hydroclimatic features for the region.

**2.2. Hydroclimate of the South American Altiplano**

Precipitation in the South American Altiplano results from the complex interactions between the large-scale atmospheric circulation and local orographic effects imposed by the topography of the Andes (Garreaud and Aceituno, 2001, Garreaud et al., 2003, Vuille and Keimig, 2004). Precipitation in the Altiplano is episodic and convective in nature, related to the zonal flow of upper air circulation from the Amazon basin (Pacific Ocean) that favors the occurrence of humid (dry) events (Garreaud, 1999, Falvey and Garreaud, 2005). More than 60% of the total annual rainfall occurs during the summer months (December-February; Vuille and Keimig, 2004). This strong seasonality is

associated with the development of the South American summer monsoon (SASM) over the tropical sector of the continent east of the Andes. One of the most prominent components of this monsoon system is an upper-level pressure cell, known as the Bolivian High, which develops over the central Andes in response to latent heat released by deep summer convection over the Amazon basin (Lenters and Cook, 1997). Moist events are related to a southward displacement of the Bolivian High, which allows the expansion of the airflow from the east and the influx of humid air masses over the Altiplano (Lenters and Cook, 1997; Garreaud et al., 2009). The zonal wind in the upper troposphere over the Altiplano is largely modulated by sea surface temperature (SSTs) across the tropical Pacific Ocean (Vuille et al., 2000; Garreaud and Aceituno, 2001; Bradley et al., 2003). Therefore, the amount of moisture reaching the Altiplano from the Amazon basin is largely modulated by the dynamics of the Bolivian High and the strength of the zonal winds in the upper troposphere (Lenters and Cook, 1997; Vuille, 1999; Garreaud and Aceituno, 2001; Garreaud et al., 2003; Vuille and Keimig, 2004).

*"In lines 424-427 you say that these differences in trends could be related to distinct time span, however the instrumental ppt data you are using are also showing negative trend, in opposite to the study published by Segura et al. 2020, is that correct? And in your study, the instrumental data you are also relatively short (~1970-present). Therefore I was asking about why you decided to calculate the average of the 7 stations, even when they had missing data."*

Our regional precipitation series is composed of seven records with less than 10% of missing data, which were completed following the methodology described in the "Climate data collection and analysis" section. We decided to generate a regional precipitation index because rainfall has a high spatial variability in the central Andes. Therefore, to minimize the influence of a single station and to highlight the common spatial and temporal pattern of variations, we developed a regional precipitation index. To clarify this issue, we added an new paragraph in the "Climate data collection and analysis" section. Please, see above the response to the reviewer comments *"In Methods you mention in…"*

Our instrumental data show a negative and significant trend when considering the period used to develop the reconstruction model, however we are aware that this trend is highly dependent on the short time period used highly influenced by the abundant rainfalls during the 1970's. Our conclusion on the negative precipitation trend is supported by the long-term trend observed in the reconstruction, not by the instrumental data as we were aware of their limitations.

*"Overall, I think this is a great manuscript, the analysis and methodology were well performed. I raised a few questions to you here that I hope can help this narrative. I do think it would be ideal to include more explanation regarding the hydroclimatology in the area, like where does the water come from and the possible explanations for this drying trend."*

We are very grateful for the time you have taken to review our manuscript and improve it through your comments.

Following your suggestions we have added a new section in Methods where we describe the hydroclimatology of the South American Altiplano (please, see response to comment *"In paragraph 409-416…).*
Also, we added in the Discussion section possible explanations for the recorded drying trend.

Since the mid-1970s, precipitation in the northern Altiplano has recorded a sustained negative trend, with the period 1997-2013 being the driest in the last 389 years. Consistent with this pattern, reconstructions of south-central Altiplano rainfall, lake size in the southwest, and mean scPDSI for the entire Altiplano also highlight the occurrence of this extreme arid event during the late 20th-early 21st century (Fig. 7). These negative trends in the hydroclimate were concurrent with the rapid retreat of glaciers across the tropical Andes during the second half of the 20th century (Ramírez et al., 2001; Francou et al., 2003; Vuille et al., 2008; Jomelli et al., 2009). In addition, this abrupt change towards more arid conditions was also recorded in the Quelccaya ice core from the 1970s onwards (Thompson et al., 2006), the Pumacocha sediment (Bird et al., 2011) and the Paleo Hydrodynamics Data Assimilation (PHYDA; Steiger et al., 2018) for the central-south Altiplano (20º-23º S, 66º-68.5º W; Fig. 8). In the context of the last four centuries of hydroclimatic variability provided by our reconstruction, the severe aridity conditions since the late 20th century are unprecedented. Due to its large spatial extent and temporal synchronicity, this extraordinary dry event across the Altiplano appears to be modulated by the action of large-scale atmospheric and oceanic forcings common to the entire region.
Previous studies show that the intensity of the summer (DJF) mean zonal winds at 200 hPa are negatively related to precipitation in the Altiplano (Minvielle and Garreaud, 2011; Neukom et al., 2015; Díaz and Vera, 2017; Morales et al. 2018). The recent increase in aridity conditions recorded in the paleorecords from the southern tropical Andes is in agreement with these studies showing a strengthening of upper tropospheric westerly winds during the last decades of the 20th century and early 21st century. Modelling exercises by Neukom et al. (2015) project an intensification of the westerlies during the 21st century, which in turn would be strongly associated with a decrease in precipitation over the Central Andes.
Li et al. (2013) indicate that ENSO activity at the end of the 20th century was anomalously high compared to the previous seven centuries. It is documented that ENSO strongly influences circulation patterns over the study area (Vuille et al., 2000, Garreaud and Aceituno, 2001), so an increase in ENSO activity would lead to stronger westerly zonal winds and less moisture input from the eastern tropical lowlands, generating drier conditions in the Central Andes. Variations in rainfall in the Altiplano are also related to changes in SAMS (Vera et al., 2006). In turn, based on high-resolution isotopic proxies, Vuille et al. (2012) associate drought conditions in the Altiplano during the last decades with a weakening of SAMS.

*"Below are a few typos I noticed."*

*Small changes/typos*
*Introduction:*
*Line 68: "found"*
*Line 73: I think you mean to say "by the end of the 2st century"*
*Line 78-79: I would suggest you only say "changes" instead of "possible current changes"*

*Line 102: "…the minimum temperature is…"*
*In "2.7 ENSO signals in the northern Altiplano precipitation" there is a typo when you write NDJ. Right now is written as "NJD"*
*Discussion*
*Line 331: "allows"*

All typos have been corrected.

---

## Author Response (AR1)

Dear editor Keely Mills,

Please, find below our responses to the reviewer comments on our submission to *Climate of the Past*, entitled "Drought increased since the mid-20$^{th}$ century in the northern South American Altiplano revealed by a 389-year precipitation record". We have prepared a new version of the manuscript that addresses most of the comments from the reviewer. In preparing this new version, we edited, rephrased and split long sentences to be shorter and clearer and we have included the two figures that were in the SI to the main text. Figure S1 is now part of Figure 2, while Figure S2 is now Figure 8. Following these changes we added new text in Method and Results to describe the figures that were not previously done in the SI.

Reviewer comments are *italicized* and in quotes, our responses are in blue, original text are in black and any substantive changes to the manuscript text are in red for reviewer #1 and in green for reviewer #2.

We thank the reviewers for their thoughtful critiques and his time and consideration.

Kind Regards,
Mariano Morales (corresponding author)

**Reviewer #1**

General comments

*"I understand that your main point is to defend the idea of an increased dryness conditions in your study area, based on a tree-ring precipitation reconstruction, however I'm not completely sold on this idea, and you show a brief discussion about it at the end of your discussion in which you mention there are studies (one at least), that show an opposite trend for the same time period, over your study area.*
*I think one of the key messages would be why would we value the tree-ring based reconstruction of precipitation you are offering over the instrumental data that are currently available. You mention the other paleoclimate proxies, however they have problems like trends that are difficult to remove sometimes."*

We consider that understanding the variability (intensity and frequency) of extreme hydroclimatic events in the Altiplano is a current topic of debate. The particular geographical location and topographic complexity of the Altiplano creates contrasting climatic conditions throughout the region. This enormous environmental complexity induces contrasting climatic trends in relatively close sites, thus limiting a solid understanding of the Altiplano's hydroclimatic variability. The low geographic density, the short-term nature of the records, frequently accompanied by the poor quality of the meteorological measurements, prevent us from capturing the complex hydroclimatic variability in the Altiplano, providing in consequence, a limited view of the mean fields and extremes of the natural hydroclimatic variability.
Although numerical and statistical approaches have improved regional future projections, the sign of future trends in water availability still remains difficult to estimate. Therefore, we believe that a long-term perspective of natural hydroclimatic variability based on climate proxies is needed to complement and contextualize present instrumental data. The driest conditions recorded in our reconstruction during the last four-five decades are also presented in independent hydroclimatic reconstructions

developed for the Andean domain. Consistent with our results, the sharp shrinkage -and in some cases the disappearance- of tropical glaciers reflects this increase in aridity conditions since the middle of the 20th century.
We have included this point in the Discussion section (line 402 to 413) and added an additional paragraph in which we put forward different hypotheses about the causes of aridization in the Central Andes (line 414-425).

*Abstract:*
*"Lines 35-40: I missed a little explanation of what would be the influences of ENSO in the precipitation in your study area."*

We have added an additional sentence to the "Abstract" to expand upon this point.

The rainfall reconstruction provides valuable information about the El Niño Southern Oscilation (ENSO) influence on the northern Altiplano precipitation, where past drier conditions in our study region were associated with ENSO warm events. The spectral properties of the rainfall reconstruction showed strong imprints of ENSO variability at decadal, sub-decadal and inter-annual time-scales, in particular from the Pacific Niño 3 sector.

*"In Methods you mention in "Climate data collection and analysis" that you used stations with less than 10% missing data. Why did you make the average of the 7 stations? Did you analyze each station against your chronologies? Or against your regional chronology?"*

To be clearer we have reworded the text describing the steps to create the regional precipitation index and added an additional paragraph to the "Climate data collection and analysis" section.

In semi-arid environments with complex topography, such as the central Andes, precipitation exhibits considerable spatial variability. In the northern Altiplano there are few instrumental records prior to 1970 and they are not regularly distributed. Therefore, to minimize the influence of any single station record and to highlight the common pattern of spatial and temporal variation across the study region, we developed a regional precipitation index. Monthly precipitation records were collected from seven meteorological stations located above 3200 m a.s.l. in the northern Altiplano between 16° and 19° S. Only station records with less than 10% of missing data were considered (Table 2). Data gaps were estimated using linear regressions (Ramos-Calzado et al., 2008). Correlation coefficients between the standard P. tarapacana regional chronology and precipitation monthly variations from each of the seven meteorological stations were used to determine the seasonal precipitation period best related to radial growth, which was from November to January (NDJ, late spring to mid-summer). Inter-annual time series of the total NDJ precipitation were computed for each of the seven available stations, normalized (z-score; Jones and Hulme, 1996) and averaged to obtain a regional index of precipitation from the northern South American Altiplano for the period 1970-2019. To assess the partial influence of temperature on tree growth we compiled monthly temperature data from nine high-altitude (>3200 m a.s.l.) meteorological stations from the Altiplano (Table 2). Based on these data, we computed a regional mean NDJ temperature index for the period 1963-2014 across the study region, following the same procedure described for developing the regional precipitation index.

*Did you try gridded data too?*

Yes, we did. However the strongest precipitation - ring width relationship was observed with instrumental records from weather stations and not with gridded products. Gridded precipitation data are spatially interpolated from instrumental data from local weather stations, but also often include remote stations, so the rainfall signal in tree rings could be relatively lower. Notwithstanding, the GPCP 2.5º x 2.5º gridded product was used to determine the spatial representativeness of our reconstruction and the instrumental regional precipitation record (please, see Figure 5a and b).

*"In line 197, did you try monthly correlations too? I'm confused, did you calculate the average of the entire year?"*

We have added a sentence that explains more clearly the process of selecting the SST months best correlated with the rainfall reconstruction. From "ENSO signals in the northern Altiplano precipitation", first paragraph:

To assess the influence of the tropical Pacific sea surface temperatures (SSTs) on the reconstructed NDJ precipitation variability over the northern Altiplano, we computed spatial correlations maps between the reconstructed NDJ precipitation and the SSTs from the gridded NCEP reanalysis global database (2.5º x 2.5º grid point resolution; Kistler et al., 2001) for the period 1949-2013. Here we test the relationships between the reconstruction and the SSTs on a monthly basis and for different sets of SSTs months: NDJ, DJF, NDJFM, Jul-Jun, Oct-Set. We found the strongest spatial correlation for the Oct-Set SST period. Since the SST shows a strong intra-annual persistence, it is expected to find a SST signal covering the whole year and not exclusively during the NDJ months corresponding to the precipitation reconstruction.

*"In line 221, is this your best correlation result? What period is this correlation?"*

Yes, this is the best correlation result for both precipitation and temperature for the common period 1970-2010, indicated in line 250-251.

*"In line 253 you say "stable", however I think you are comparing this ~ 20 years frequency with the increased frequency you observe in the 20-21th centuries, is that correct? Stable to me sounds like nothing was happening, however it was, but at a lower frequency. I suggest you change the text in this paragraph, there is no such a thing as stable in climate, because it is constantly changing."*

In this paragraph we use the word "stable" to refer to the fact that the recurrence time of extreme events remains relatively low and constant during the 18th and 19th centuries. In accordance with the reviewer's comment, we have eliminated "stable" and reworded the paragraph.

The 18th and 19th centuries showed a relatively low and constant recurrence rate of extreme dry events with a drought every ~20-27 years, while from 1940 to 2013, there was a steady increase in extreme dry events from ~17 to ~9 years (Fig. 4b).

*"In lines 345-347 I ask myself, what are the odds of your chronologies showing decreased growth due to increased temperature and not exactly related to decrease in ppt?"*

In lines 364-378 we state that the rainfall reconstruction model for the northern Altiplano region explained a lower percentage of total variance with respect to other reconstructions developed in the Altiplano and we put forward some alternative hypotheses about the possible causes of this difference in the R2 values.

Our model explains 42.5% of the total rainfall variations, a relatively lower percentage than expected by other tree-ring based hydroclimate reconstructions for the Altiplano. Therefore, there are additional factors related to tree growth and meteorological precipitation information that could be affecting the skill of our model, such as:

1- The low number, quality, spatial coverage and brevity of the instrumental records used to develop the reconstruction model.

2- For instance, NDJ precipitation is the best-correlated period with tree-growth, but other months also influence growth in a less proportion.

3- The lower number of tree samples used to generate the northern Altiplano chronology compared to the south-central reconstruction.

4- Differences in hydroclimate conditions prevailing throughout the entire Altiplano and/or species sensitivity to precipitation should also be considered as potential causes for the observed difference in R2

5- There are others climatic variables such as temperature and wind that contributes to increase evapotranspiration from plants and soil evaporation demand, generating more severe dry conditions. Trees integrate the responses to these biotic and abiotic factors in their growth. However, using dendrochronological methods we can determine which of these factors is most closely related to interannual growth variations. In the Altiplano, rainfall plays a major role in tree growth, as we have already documented in previous work (Morales et al. 2004; 2012; Solíz et al. 2009; Rodriguez-Catón et al. 2021; Crispín De La Cruz et al. 2022). In the Altiplano, precipitation and temperature co-vary inversely: radiation is higher on clear days without convection inducing an increase in temperature, which in turn leads to higher evapotranspiration and a decrease in water availability affecting tree growth.

Finally, the comparison with other independent paleoclimatic records of precipitation in the Altiplano is important to reduce uncertainties. In Figure 7, we show the good relationships between the precipitation reconstructions developed for the Central and Southern Altiplano with our reconstruction for the northern sector.

We have reworded the second paragraph of the discussion, mentioned other factors such as the quality of meteorological records that could contribute to the recorded R2 differences among the different reconstructions across the Altiplano. In addition, we have reduced the emphasis on secondary factors associated with tree growth that may affect the strength of relationship between growth and precipitation.

The regression model used to develop the reconstruction explains 42.5% of the total variance of the late spring (November) to the mid summer (January) precipitation for the 1970-2010 calibration period. This value is lower than that previously reported for a tree-ring based precipitation reconstruction in the central-southern Altiplano ($R^2 = 0.56$; Morales et al., 2012). A lower sample size of trees used to generate the northern regional chronology in comparison with the southern-central regional chronology could have influenced the relatively lower correlation obtained and highlights the importance

of fieldwork expeditions to increase the number of centennial tree-ring chronologies in the northern sector of the Altiplano. The poor quality, low spatial coverage and brevity of the climate instrumental records, could also affect the climate-growth relationships used to develop the reconstruction model. Other factors such as differences in hydroclimate conditions prevailing throughout the entire Altiplano and/or species sensitivity to precipitation should also be considered as potential causes for the observed difference in $R^2$. In the northern Altiplano, rainfall is almost double that recorded in the southern region (i.e. 300-400 mm vs. 100-200 mm, respectively; Crispín-DeLa Cruz et al., 2022). Therefore, precipitation at our study site would be less limiting for tree-growth than in the drier southern regions of the Altiplano (Solíz et al., 2009; Rodríguez-Catón et al., 2021). While rainfall explains a high percentage of the total variance of radial growth, temperature variations could also play a secondary role via evapotranspiration, which regulate stomatal aperture and carbon fixation for *P. tarapacana* growth as suggested by Rodríguez-Catón et al. (2021).

*"In line 370 you mention 1876/1877, and according to your dated chronology, your 1876 ring is the one that started in November of 1876 and extended to 1877. That would be already when the El Nino happened, and yet you say your 1876 shows as the wettest year."*

Thank you very much for pointing out this dating error. The Niño event occurred during calendar years 1877/78 (Singh et al. 2018) and not in 1876/77 as stated in the original text. Therefore, the occurrence of the 1877/78 Niño event coincides with the occurrence of an extreme drought event recorded in the 1877 year of our reconstruction. The year 1877 of our reconstruction represents the months corresponding to November and December 1877 and January 1878.

We have changed the date in the manuscript, line 400.

*"In line 385 I would suggest you write what BT stands for, I know you already showed in methods, but it's a long way for the readers to go back and find the abbreviation."*

We added the complete name for the Blackman-Tukey analysis.

*"In paragraph 409-416: What is the main source of precipitation in the study area? Is it the Pacific Ocean or the glaciers? Is this drying trend related to losses in ice cover, increased temperature, or both, and are you suggesting increased SST's activity?"*

In line 55 of the Introduction we briefly describe the main source of precipitation in the Altiplano. However, we considered pertinent the reviewer's suggestion and decided to incorporate a new section in Materials and Methods describing the main hydroclimatic features for the region.

**2.1. Precipitation of the South American Altiplano**

Precipitation in the South American Altiplano results from the complex interactions between the large-scale atmospheric circulation and local orographic effects imposed by the topography of the Andes (Garreaud and Aceituno, 2001, Garreaud et al., 2003; Vuille and Keimig, 2004). Altiplano precipitation is episodic and convective in nature,

related to the zonal flow of upper air circulation from the Amazon basin (Pacific Ocean) that favors the occurrence of humid (dry) events (Garreaud, 1999; Falvey and Garreaud, 2005). More than 60% of the total annual rainfall occurs during the summer months (December-February; Vuille and Keimig, 2004). This strong seasonality is associated with the development of the South American summer monsoon (SASM) over the tropical sector of the continent east of the Andes. The SASM presents a seasonal cycle including the onset (October-November), maturity (December-February) and die-off (March-April) periods. One of the most prominent components of this monsoon system is an upper-level pressure cell, known as the Bolivian High, which develops over the central Andes in response to latent heat released by deep summer convection over the Amazon basin (Lenters and Cook, 1997). Moist events are related to a southward displacement of the Bolivian High, which allows the expansion of the airflow from the east and the influx of humid air masses over the Altiplano (Lenters and Cook, 1997; Garreaud et al., 2009). The zonal wind in the upper troposphere over the Altiplano is largely modulated by sea surface temperature (SSTs) across the tropical Pacific Ocean (Vuille et al., 2000; Garreaud and Aceituno, 2001; Bradley et al., 2003). Therefore, the amount of moisture reaching the Altiplano from the Amazon basin is largely modulated by the dynamics of the Bolivian High and the strength of the zonal winds in the upper troposphere (Lenters and Cook, 1997; Vuille, 1999; Garreaud and Aceituno, 2001; Garreaud et al., 2003; Vuille and Keimig, 2004).

*"In lines 424-427 you say that these differences in trends could be related to distinct time span, however the instrumental ppt data you are using are also showing negative trend, in opposite to the study published by Segura et al. 2020, is that correct? And in your study, the instrumental data you are also relatively short (~1970-present). Therefore I was asking about why you decided to calculate the average of the 7 stations, even when they had missing data."*

Our regional precipitation series is composed of seven records with less than 10% of missing data, which were completed following the methodology described in the "Climate data collection and analysis" section. We decided to generate a regional precipitation index because rainfall has a high spatial variability in the central Andes. Therefore, to minimize the influence of a single station and to highlight the common spatial and temporal pattern of variations, we developed a regional precipitation index. To clarify this issue, we added an new paragraph in the "Climate data collection and analysis" section. Please, see above the response to the reviewer comments "*In Methods you mention in…*"

Our instrumental data show a negative and significant trend when considering the period used to develop the reconstruction model, however we are aware that this trend is highly dependent on the short time period used highly influenced by the abundant rainfalls during the 1970's. Our conclusion on the negative precipitation trend is supported by the long-term trend observed in the reconstruction, not by the instrumental data as we were aware of their limitations.

*"Overall, I think this is a great manuscript, the analysis and methodology were well performed. I raised a few questions to you here that I hope can help this narrative. I do think it would be ideal to include more explanation regarding the hydroclimatology in the area, like where does the water come from and the possible explanations for this drying trend."*

We are very grateful for the time you have taken to review our manuscript and improve it through your comments.
Following your suggestions we have added a new section in Methods where we describe the hydroclimatology of the South American Altiplano (please, see response to comment *"In paragraph 409-416…)*.
Also, we added in the Discussion section possible explanations for the recorded drying trend.

In the context of the last four centuries of hydroclimatic variability provided by our reconstruction, the severe aridity conditions since the late 20th century are unprecedented. Due to its large spatial extent and temporal synchronicity, this extraordinary dry event across the Altiplano may be modulated by the action of large-scale atmospheric and oceanic forcings common to the entire region such as zonal wind and ENSO variability. Previous studies show that the intensity of the summer (DJF) mean zonal winds at 200 hPa are negatively related to precipitation in the Altiplano (Minvielle and Garreaud, 2011; Neukom et al., 2015; Díaz and Vera, 2018; Morales et al. 2018). The recent increase in aridity conditions recorded in the paleorecords from the southern tropical Andes is in agreement with these studies showing a strengthening of upper tropospheric westerly winds during the last decades of the 20th century and early 21st century. Modelling exercises by Neukom et al. (2015) project an intensification of the westerlies during the 21st century, which in turn would be strongly associated with a decrease in precipitation over the Central Andes.
Regarding ENSO, Li et al. (2013) reported an activity increase at the end of the 20th century that was anomalously high compared to the previous seven centuries. It is documented that ENSO strongly influences circulation patterns over the study area (Vuille et al., 2000, Garreaud and Aceituno, 2001), thus an increase in ENSO activity could lead to stronger westerly zonal winds and less moisture input from the eastern tropical lowlands, generating drier conditions in the Central Andes. Variations in rainfall in the Altiplano are also related to changes in SAMS (Vera et al., 2006) as illustrated by high-resolution isotopic proxies that reported drought conditions in the Altiplano associated to a weakening of SAMS during the last decades (Vuille et al. 2012).

*"Below are a few typos I noticed."*

*Small changes/typos*
*Introduction:*
*Line 68: "found"*
*Line 73: I think you mean to say "by the end of the 2st century"*
*Line 78-79: I would suggest you only say "changes" instead of "possible current changes"*
*Line 102: "...the minimum temperature is…"*
*In "2.7 ENSO signals in the northern Altiplano precipitation" there is a typo when you write NDJ. Right now is written as "NJD"*
*Discussion*
*Line 331: "allows"*

All typos have been corrected.

**Reviewer #2**

We thank the reviewer for their thoughtful critiques and his time and consideration.

General comments

*"The authors often cite the issue of water stress and how this can inform management - and this is alluded to in the text at several points. It could be useful for the authors to articulate exactly how their study can be used in management. It would provide context, but as written can not inform management of water supplies. Perhaps more articulation (in a few sentences) could be included in the discussion section of this paper. Overall the quality of the science and the manuscript is very high. There are a few areas where sentences are a little long, and need to be shortened and/or rephrased to make sure the information is not lost. I attach an annotated manuscript with corrections / edits for the authors to undertake."*

To be more specific about the importance of paleoclimatic records as a source of information potentially useful for decision-makers, we rephrase the following paragraph in the Discussion section:

Droughts are of particular relevance in climate variability for this semi-arid region of the Andes. Therefore, the information provided by this study allows us to understand that mean dry condition dominates the instrumental period and the frequency of occurrence of extreme drought events in the present has no precedent in the past. Under a global warming context, the Altiplano's water resources are fundamental for biodiversity conservation and socioeconomic activities. The projected increase in evapotranspiration as a result of global warming, together with a wide range of variability among the precipitation models projected for the 21st century, may lead to growing demand for water in a region already under water stress. Knowing the long-term hydroclimatic variability in this region, we need to consider whether the current configuration of social and organizational structures are sufficient to provide the resilience and adaptation to successfully address current and future hydroclimatic changes. A better understanding of the future of Altiplano's water resources should be listed as priority for stakeholders and decision-makers to avoid social conflicts at both the local and regional levels. Under this complex political, social and environmental scenario, the results from our study are relevant to plan and implement adaptive strategies to reduce these vulnerabilities in the face of future water shortages.

Specific comments

*Line 24. "There are several very long sentences in this abstract. I have tried to split some where possible. Please check that sentences are a little bit shorter - it makes it easier to read."*

The abstract and manuscript were revised in order to shorten, where possible, very long sentences.

*Line 29. "CLARIFY - this would need rephrasing - it doesn't quite make sense.*

*What gap? Temporal? Spatial? In our knowledge of hydroclimatic change?"*

To be clearer, we rephrase the following sentence:

However, in the northern sector of the Peruvian and Chilean Altiplano (16º-19º S) still exist a gap of hydroclimatic tree-ring records.

By

However, in the northern sector of the Peruvian and Chilean Altiplano (16º-19º S) still exist a gap in our knowledge of high-resolution hydroclimatic change based on tree-ring records.

*Line 99 – 102. Do you mean >85% of the rainfall occurs from December to March? If so, you need to clarify this. If not, then it will need to be clarified as I'm not sure I understand.*
*"Can you give actual numbers here - this isn't the best phrasing and has also appeared immediately before"*

Thanks for the comment. We rephrase and split the sentence to be shorter and clearer.

The climate is semi-arid with dry-cold winters and rainy-warm summers, with a total annual precipitation that ranges between 290-400 mm. More than 85% of the total annual precipitation occurs during summer (December to March).

*Line 300-302. "This description doesn't make sense. Can you rephrase"*

To be clearer, we rephrase the following sentence:

Strong correlations (r > -0.42) were recorded centered in the periods 1885-1900, 1940-1955, 1965-1975, while sharp decreases in correlations were found centered in the periods 1910-1935, 1978-2000 (Fig. 6c).

by

Strong negative correlations were recorded centered in the periods 1885-1900, 1940-1955, 1965-1975, while a lost in the relationships was found centered in the periods 1910-1935, 1978-2000 (Fig. 6c).

*Line 303-307. "Not sure this makes sense. What explains the high percentage. Need to split sentence into 2 and clarify."*

Both, reconstructed precipitation and SSTs_N3 recorded dominant oscillations modes at decadal (Fig. 6d) and inter-annual (Fig. 6e,f) frequencies explaining high percentage of their variability. A correlation analyses among the main dominant oscillatory modes at decadal 10.2 year (Fig. 6d) and inter-annual frequencies 6.2 year (Fig. 6e) and 3.5 year (Fig. 6f) of the precipitation reconstruction and the SSTs_N3 with 12.7 year, 5.6 year and 3.6 year, respectively, showed in the three cases significant negative correlations.

The sentence was clarify as follow:

High percentage of the variability in the recorded precipitation reconstruction and the SSTs_N3 is dominated by oscillations modes at decadal (Fig. 6d) and inter-annual (Fig. 6e,f) frequencies. Correlation analyses among the main dominant oscillatory modes of the precipitation reconstruction and the SSTs_N3 at decadal (Fig. 6d) and inter-annual (Fig. 6e,f) ferquencies, showed significant negative correlations. These results highlight the occurrence of common waveforms and demonstrate the ENSO signal in the precipitation reconstruction from the northern South American Altiplano.

*Line 367-371. "The phrasing of this doesn't make sense. please clarify"*

The wettest year recorded in our reconstruction is 1876, which is associated with the occurrence of a prolonged cool phase of the central tropical Pacific during 1870–1876 period (La Niña conditions), being the year 1876 the coldest SST record (Singh et al., 2018). This cool La Niña phase conditions reversed to a warm SSTs during the strong 1876/1877 El Niño event (Singh et al., 2018), which is registered in our precipitation reconstruction as an extreme dry year in 1877.

There was a dating error. The Niño event occurred during calendar years 1877/78 (Singh et al. 2018) and not in 1876/77 as stated in the original text. Therefore, the occurrence of the 1877/78 Niño event coincides with the occurrence of an extreme drought event recorded in the 1877 year of our reconstruction. The year 1877 of our reconstruction represents the months corresponding to November and December 1877 and January 1878.

We have changed the date in the new version of the manuscript

*Line 380-383. "Can you give actual numbers here - this isn't the best phrasing and has also appeared immediately before".*

The following sentence was rephrased for clarity

It is important to place this increase in aridity conditions since the late 20th century - beginning 21st century observed in our northern record in the long-term context and the great spatial coherence and synchrony shown by all the other proxies records across the southern tropical Andes (Fig. 7 and Fig. S2), suggesting largescale common atmospheric and ocean forcings over this Andean region

By

In the context of the last four centuries of hydroclimatic variability provided by our reconstruction, the severe aridity conditions since the late 20th century are unprecedented. Due to its large spatial extent and temporal synchronicity, this extraordinary dry event across the Altiplano may be modulated by the action of large-scale atmospheric and oceanic forcings common to the entire region

*Line 419-422. "Sentence too long and too complex. Please rephrase and clarify."*

Segura et al. (2020) based on instrumental-satellite precipitation data for the southern region of the tropical Andes (12°-20° S; 60°-80° W), evaluate the common pattern of

summer rainfall variation for this region during the period 1982-2018, identifying a positive trend specially based in positive anomalies after 2010 that would be influenced by upward motion over the western Amazon.

To be clearer and shorter, we split the sentence as follow:

Segura et al. (2020) based on instrumental-satellite precipitation data for the southern region of the tropical Andes (12º-20º S; 60º-80º W), evaluate the common pattern of summer rainfall variation for this region during the period 1982-2018. They identified a positive trend especially after 2010, which would be influenced by upward motion over the western Amazon.

All typos have been corrected in the new version of our manuscript.